# Achievable distributional robustness when the robust risk is only partially identified

**Julia Kostin**
Department of Computer Science
ETH Zurich
jkostin@ethz.ch

**Nicola Gnecco**
Gatsby Computational Neuroscience Unit
University College London
nicola.gnecco@gmail.com

**Fanny Yang**
Department of Computer Science
ETH Zurich
fan.yang@inf.ethz.ch

## Abstract

In safety-critical applications, machine learning models should generalize well under worst-case distribution shifts, that is, have a small robust risk. Invariance-based algorithms can provably take advantage of structural assumptions on the shifts when the training distributions are heterogeneous enough to identify the robust risk. However, in practice, such identifiability conditions are rarely satisfied – a scenario so far underexplored in the theoretical literature. In this paper, we aim to fill the gap and propose to study the more general setting of *partially identifiable robustness*. In particular, we define a new risk measure, the worst-case robust risk, and its corresponding (population) minimax quantity that is an algorithm-independent measure for the best achievable robustness under partial identifiability. We introduce these concepts broadly, and then study them within the framework of linear structural causal models for concreteness of the presentation. We use the introduced minimax quantity to show how previous approaches provably achieve suboptimal robustness in the partially identifiable case. We confirm our findings through empirical simulations and real-world experiments and demonstrate how the test error of existing robustness methods grows increasingly suboptimal as the proportion of previously unseen test directions increases.

## 1   Introduction

The success of machine learning methods typically relies on the assumption that the training and test data follow the same distribution. However, this assumption is often violated in practice. For instance, this can happen if the test data are collected at a later time or using a different measuring device. Without further assumptions on the test distribution, generalization under distribution shift is impossible. However, practitioners often have partial information about the set of possible "shifts" that may occur during test time, inducing a set of *feasible test distributions* that the model should generalize to. We refer to the resulting set as the *robustness set*. When a probabilistic model for these possible test distributions is available or estimable, one may aim for good performance on a "typical" held-out distribution using a probabilistic framework. When no extra information is available or estimable, one possibility is to find a model $\beta$ that has a small risk $\mathcal{R}(\beta; \mathbb{P})$ on the hardest feasible test distribution. More formally, we aim to achieve a small robust risk defined by

$$\mathcal{R}_{\mathrm{rob}}(\beta) := \sup_{\mathbb{P} \in \mathcal{P}_{\mathrm{rob}}(\theta^\star)} \mathcal{R}(\beta; \mathbb{P}), \tag{1}$$

38th Conference on Neural Information Processing Systems (NeurIPS 2024).

Table 1: Comparison of various distributional robustness frameworks and what kind of assumptions their analysis can account for (with an incomplete list of examples for each framework).

| Framework accounts for | bounded shifts | partial identifiability of model parameters | partial identifiability of robustness set |
|---|---|---|---|
| DRO [7, 15, 49, 32, 43] | ✓ | – | ✗ |
| Infinite robustness [35, 17, 30, 39, 6, 2, 46, 54, 28, 1] | ✗ | ✗ | ✗ |
| Finite robustness [41, 23, 14, 27, 45] | ✓ | ✓ | ✗ |
| Partially id. robustness (this work) | ✓ | ✓ | ✓ |

where $\mathcal{P}_{\mathrm{rob}}(\theta^\star)$ corresponds to the robustness set which depends on some true parameter $\theta^\star$. In fact, this worst-case robustness aligns with security and safety-critical applications, where a small robust risk is necessary to confidently guarding against possible malicious attacks.

Causality-oriented robustness [11, 31, 45] on the other hand is based on the idea that some structural parameters (like a graphical structure of the model) remain invariant across distributions, while others may vary. For a given set of training distributions, certain sets of varying parameters induce robust risks that are identifiable. Similarly, for a given set of varying parameters, heterogeneous enough training distributions may identify the robust risk.

In practice, robustness methods aiming at minimizing a pre-defined robust risk often suffer from ineffectiveness. For adversarial robustness for example, it is known that when the perturbations during training and test time differ, the robust risks resulting from adversarial training and standard training may be comparable (see, e.g. [52, 26]). Similarly, invariance-based methods are often shown to be less robust than vanilla empirical risk minimization that ignores multi-environment information. Theoretically, besides being effective only for very specific data-generating models [1], invariance-based methods generally are bound to fail when the heterogeneity of the training data is not enough for a given set of possible test shifts. Even though this issue of non-identifiability has been pointed out previously [25, 40], prior work so far was primarily satisfied with such a binary statement - whether identifiability is given or not. We believe that the non-identifiable scenario warrants a more detailed discussion. In particular, we aim to formalize how to quantify the best possible robustness for this partially identifiable setting. In particular, we extend the discussion of invariance-based methods to include the partially identifiable setting, where not only the causal parameter, but the robust risk (1) is not determinable using training data either[1]. Specifically, we aim to discuss the following question:

*What is the optimal worst-case performance any model can have for given structural relationships between test and training data and how do existing methods comparatively perform in such settings?*

When the robust risk is not identifiable from training data, we obtain a whole *set* of possible objectives that includes the true robust risk. In this case, we are interested in the best achievable robustness for *any algorithm* that we capture in a quantity called the *worst-case robust risk*:

$$\mathfrak{R}_{\mathrm{rob}}(\beta) := \sup_{\substack{\text{possible} \\ \text{true model } \theta^\star}} \sup_{\mathbb{P} \in \mathcal{P}_{\mathrm{rob}}(\theta^\star)} \mathcal{R}(\beta; \mathbb{P}). \tag{2}$$

Note that $\mathfrak{R}_{\mathrm{rob}}(\beta)$ is well-defined even when the standard robust risk is not identifiable – it takes the supremum over the robust risks induced by all possible true model parameters $\theta^\star$ that are consistent with the given set of training data distributions. Furthermore, the minimal value of the identifiable robust risk corresponds to the optimal worst-case performance in the partially identifiable setting. Spiritually, this *minimax population quantity* is reminiscent of the algorithm-independent limits in classical statistical learning theory [55].[2] Even though our partial identifiability framework can be

---

[1]Here, we mean partial identifiability of the robust risk, which is reminiscent of outputting uncertainty sets for a quantity of interest in the field of partial identification [50, 18].

[2]In particular, extending (2) to its finite-sample counterpart would introduce a more natural extension of the classical minimax risk statistical learning theory. In this work, we focus on identifiability aspects instead of statistical rates.

evaluated for arbitrary modeling assumptions on the distribution shift (such as covariate/label shift, DRO, etc.), we present it in a concrete linear setting for clarity of the exposition. Specifically, the setting is motivated by structural causal models (SCMs) with unobserved confounding (cf. Section 2), similar to the setting of IV (instrumental variables) and anchor regression [41, 42]. Concurrently with our work, [24] proposed a framework for partial transportability which is conceptually related to our notion of worst-case robust risk. However, their approach leverages graphical assumptions, i.e., a priori knowledge about the structure of causal models, whereas our focus is a more agnostic multi-environment setting. Additionally, we do not assume a causal data generation process.

The worst-case robust risk (2) not only represents a notion of algorithm-independent optimality for any combination of training and test shifts. In the linear setting in Section 2, we also show theoretically and empirically that the ranking and optimality of different robustness methods change drastically in identifiable vs. partially identifiable settings. The same can be observed in experiments on real-world data. Our experimental results strongly indicate that evaluation and benchmarking on partially identifiable settings are important for determining the effectiveness of robustness methods. Finally, while the worst-case robust predictor derived in the paper is only provably optimal for the linear setting, experiments on real-world data in Section 4 suggest that our estimator may significantly improve upon other invariance-based methods in more realistic scenarios.

## 2 Setting

In this section, we state the concrete distributional setting on which we introduce our partial identifiability framework. In particular, we consider a data generating process, motivated by structural causal models (SCMs), that allows for hidden confounding, i.e., spurious correlations between the covariates $X$ and the target $Y$. We describe the structure of the distribution shifts occurring in the training and test environments, which is reminiscent of interventions in causal models. Finally, we introduce our framework for distributional robustness that allows for partial identifiability and define the *worst-case robust risk* – for any given model, it corresponds to the maximum robust risk among all possible robust risks induced by the training distributions.

### 2.1 Data distribution and a model of additive environmental shifts

**Data generating process (DGP).** We first describe the data-generating mechanism that underlies the distributions of all environments $e \in \mathcal{E}$ that may occur during train or test time. For each environment $e \in \mathcal{E}$, we observe the random vector $(X_e, Y_e) \sim \mathbb{P}_e^{X,Y}$ consisting of input covariates $X_e \in \mathbb{R}^d$ and a target variable $Y_e \in \mathbb{R}$ which satisfy the following data generating process:

$$
\begin{aligned}
X_e &= A_e + \eta; \\
Y_e &= \beta^{\star\top} X_e + \xi,
\end{aligned}
\tag{3}
$$

where $A_e \in \mathbb{R}^d$, $(\eta, \xi) \in \mathbb{R}^{d+1}$ are random vectors $A_e \sim \mathbb{P}_e^A$, $(\eta, \xi) \sim \mathbb{P}^{\eta,\xi}$ with finite first and second moments and for which $A_e \perp\!\!\!\perp (\eta, \xi)$ for all $e \in \mathcal{E}$.

**Invariant mechanism.** Note how in this setting, apart from $\beta^\star$, the distribution $\mathbb{P}^{\eta,\xi}$ of the noise vector $(\eta, \xi)$ remains constant across environments. Without loss of generality, we assume that the noise $(\eta, \xi)$ has mean zero. Note how this linear setting is, in general, more challenging than the standard linear regression setting where $\eta \perp\!\!\!\perp \xi$: due to possible dependencies between $\eta$ and $\xi$ (induced by, e.g., *hidden confounding/spurious features*), classical estimators, such as the ordinary least squares, are biased away from the true parameter $\beta^\star$. Denote by $\Sigma^\star := \mathrm{Cov}((\eta, \xi))$ the joint covariance of the noise vector $(\eta, \xi)$, which can be written in block form as $\Sigma^\star = \begin{pmatrix} \Sigma_\eta^\star & \Sigma_{\eta,\xi}^\star \\ \Sigma_{\eta,\xi}^{\star\top} & (\sigma_\xi^\star)^2 \end{pmatrix}$ and which we assume to be full-rank. We then denote the concatenation of these two invariant parameters by $\theta^\star := (\beta^\star, \Sigma^\star) \in \Theta \subset \mathbb{R}^d \times \mathbb{R}^{(d+1) \times (d+1)}$ - the parameter that remains invariant across all environments.

**Structure of the distribution shifts.** Note that in the DGP, the distribution shifts between $\mathbb{P}_e^{X,Y}$ are induced solely by changes in the distribution of the variable $A_e$, whose mean and covariance matrix we denote by $\mathbb{E}[A_e] = \mu_e$ and $\mathrm{Cov}[A_e] = \Sigma_e$, respectively. In general, we allow for degenerate shifts, i.e. the covariance $\Sigma_e$ can be singular. We remark that although the additive shift structure in Equation (3) allows the joint distribution $\mathbb{P}_e^{X,Y,A} = \mathbb{P}_e^A \times \mathbb{P}^{X,Y|A}$ to change solely via $\mathbb{P}_e^A$, our distribution shift setting is more general than covariate shift: due to the noise variables $\eta$ and $\xi$ being

potentially dependent, both the marginal $\mathbb{P}_e^X$ and the conditional distribution $\mathbb{P}_e^{Y|X}$ can change across environments.

**Training and test-time environments.** Throughout the paper, we assume that we are given the collection of training distributions $\mathcal{P}_{\theta^\star, \mathcal{E}_{\text{train}}} = \{\mathbb{P}_{\theta^\star, e}^{X,Y}\}_{e \in \mathcal{E}_{\text{train}}}$, where $\mathcal{E}_{\text{train}}$ denotes the index set of training environments. We omit $\theta^\star$ when it is clear from the context. Further, for ease of exposition, we assume that $\mathcal{E}_{\text{train}}$ contains a reference (unshifted) environment $e = 0$ with $A_0 = 0$ a.s. In Appendix B, we discuss how our results apply if this condition is not met. During test time, we expect to observe a new, previously unseen distribution $\mathbb{P}_{\text{test}}^{X,Y}$ which is induced by the DGP (3) and a shift random variable $A_{\text{test}} \sim \mathbb{P}_{\text{test}}^A$, with corresponding finite mean $\mu_{\text{test}}$ and covariance $\Sigma_{\text{test}}$.

Even though we do not have access to $\mathbb{P}_{\text{test}}^{X,Y}$ during training, the practitioner might have some information about the possible shift distributions $\mathbb{P}_{\text{test}}^A$ that may occur during test time. As an example, we may only have information about the maximum possible magnitude and direction of the test-time mean shift $\mathbb{E}[A_{\text{test}}]$. In this work, we assume that we are given an upper bound on the second moment of the shift variable, represented by a positive semidefinite (PSD) matrix $M_{\text{test}} \succeq 0$ such that

$$\mathbb{E}[A_{\text{test}} A_{\text{test}}^\top] = \Sigma_{\text{test}} + \mu_{\text{test}} \mu_{\text{test}}^\top \preceq M_{\text{test}}. \tag{4}$$

In practice, there may be different degrees of knowledge of the feasible set of shifts – when no knowledge is available, one can always choose the most "conservative" bound $M_{\text{test}}$ with the range equal to $\mathbb{R}^{d \times d}$ and large eigenvalues. The more information is available, the smaller the feasible set of test distributions would become. On the other hand, when the test distribution $\mathbb{P}_{\text{test}}^X$ of $X$ is available during training (as in the *domain adaptation* setting [47]), one can directly compute the optimal shift upper bound via $M_{\text{test}} = \mathbb{E}[X^{\text{test}} X^{\text{test}\top}]$. In existing literature, $M_{\text{test}}$ is often proportional to the pooled first or second moment of the training shifts, for instance $M_{\text{test}} = \gamma \sum_{e \in E_{\text{train}}} w_e \mu_e \mu_e^\top$ in discrete anchor regression [41] or $M_{\text{test}} = \gamma \sum_{e \in E_{\text{train}}} w_e (\mu_e \mu_e^\top + \Sigma_e)$ in causality-oriented robustness with invariant gradients [45]. Here, $w_e$ are the weights representing the probability of a datapoint being sampled from the environment $e$. As will become apparent in the next sections, our population-level results are not impacted by the distribution of the environment variable, which we thus omit in the following.

We now provide an example based on structural causal models (SCM) that falls under the aforementioned distrubtion shift setting.

*Example* 1. Consider the structural causal model and its induced graph in Figure 1. In this model, the variable $Z$ is a soft intervention on the covariates $X$. Additionally, the exogenous noise vector $(\varepsilon_X, \varepsilon_Y, \varepsilon_H)$ and the intervention variable $Z$ are mutually independent. This model is the basis of multiple causality-oriented robustness works, e.g. [41, 45]. Let $\beta^\star := B_{YX}^\top$ and $\xi := B_{YH} H + \varepsilon_Y$. Then, from (5), we obtain $Y = B_{YX} X + (B_{YH} H + \varepsilon_Y) = X^\top \beta^\star + \xi$. Suppose that $\mathbf{I} - \mathbf{B}$ is invertible and let $\mathbf{C} := (\mathbf{I} - \mathbf{B})^{-1}$ with entries $C_{XX}, C_{XY}$, etc. Define $A := C_{XX} Z$ and $\eta := C_{XX} \varepsilon_X + C_{XY} \varepsilon_Y + C_{XH} \varepsilon_H$. Then, we can write $X = A + \eta$. Since shifts in distribution of $Z$ induce shifts in the distribution of $A$, a collection of interventions $\{Z_e\}_{e \in \mathcal{E}_{\text{train}}}$ translates into a collection of additive shifts $\{A_e\}_{e \in \mathcal{E}_{\text{train}}}$ and gives rise to training distributions varying with the environment $e$. In summary, our DGP Equation (3) includes the classical setting of causality-oriented robustness as depicted in Figure 1.

$$\begin{pmatrix} X \\ Y \\ H \end{pmatrix} = \underbrace{\begin{pmatrix} B_{XX} & B_{XY} & B_{XH} \\ B_{YX} & 0 & B_{YH} \\ 0 & 0 & 0 \end{pmatrix}}_{\mathbf{B}} \begin{pmatrix} X \\ Y \\ H \end{pmatrix} + \begin{pmatrix} Z + \varepsilon_X \\ \varepsilon_Y \\ \varepsilon_H \end{pmatrix} \tag{5}$$

Figure 1: (Left) SCM with hidden confounding and (Right) induced graph. The model allows for an arbitrary causal structure of the observed variables $(X, Y)$, as long as $\mathbf{I} - \mathbf{B}$ is invertible, i.e. the underlying graph is acyclic. The shifts across different distributions are captured via shift interventions on $X$, however, the model does not allow for interventions on the target variable or hidden confounders.

## 2.2 The robust risk

Our goal is to find an estimator using the training data that has a small risk, in this paper exclusively the expected square loss $\mathcal{R}(\beta; \mathbb{P}) := \mathbb{E}_\mathbb{P}[(Y - \beta^\top X)^2]$, over the robustness set. In our setting, given

a test shift upper bound $M_{\text{test}}$ defined in Equation (4), the robustness set corresponds to

$$\mathcal{P}_{\theta^\star}(M_{\text{test}}) := \{\mathbb{P}^{X,Y}_{\theta^\star,\text{test}} : \mathbb{E}\left[A_{\text{test}}A_{\text{test}}^\top\right] \preceq M_{\text{test}}\}, \tag{6}$$

yielding the corresponding robust risk that reads

$$\mathcal{R}_{\text{rob}}(\beta; \theta^\star, M_{\text{test}}) := \sup_{\mathbb{P} \in \mathcal{P}_{\theta^\star}(M_{\text{test}})} \mathcal{R}(\beta; \mathbb{P}). \tag{7}$$

We call the minimizer of the robust risk $\beta^{rob}_{\theta^\star} := \arg\min_{\beta \in \mathbb{R}^d} \mathcal{R}_{\text{rob}}(\beta; \theta^\star, M_{\text{test}})$ the *robust predictor*. For the squared loss and linear model, the robust risk can be computed in closed form and *solely* depends on $M_{\text{test}}$ and the invariant parameters $\theta^\star = (\beta^\star, \Sigma^\star)$, and not on other properties of the distributions:

$$\mathcal{R}_{\text{rob}}(\beta, \theta^\star, M_{\text{test}}) = (\beta^\star - \beta)^\top(\Sigma^\star_\eta + M_{\text{test}})(\beta^\star - \beta) + 2(\beta^\star - \beta)^\top\Sigma^\star_{\eta,\xi} + (\sigma^\star_\xi)^2. \tag{8}$$

This observation motivates us to define an equivalence relation between two data-generating processes that holds whenever they induce the same robust risk for any model $\beta$ and shift upper bound $M_{\text{test}}$. Specifically, observe that $\text{DGP}_1$ and $\text{DGP}_2$ induce the same robust risks for all $M_{\text{test}}$ and $\beta$ iff $\beta^\star_1 = \beta^\star_2$ and $\mathbb{P}^{\eta,\xi}_1 \cong \mathbb{P}^{\eta,\xi}_2$, where $\cong$ denotes the equivalence of distributions based on equality of their first and second moments. Thus, in the following, we treat our data-generating process as uniquely defined by $\theta^\star$ up to this equivalence relation.

In practice, the model parameters $\theta^\star$ typically cannot be identified from the training distributions, and the robust risk $\mathcal{R}_{\text{rob}}$ can only be computed for specific combinations of training and test shifts, studied, e.g., in [41, 45]. In the next section, we describe concepts that allow us to reason about robustness in the case when the robust risk is only partially identifiable.

## 2.3 Partially identifiable robustness framework

We start by formally introducing the general notion of partial identifiability for the robust risk. The following notion of *observational equivalence* of parameters is reminiscent of the corresponding notion in the econometrics literature [16]:

**Definition 1** (Observational equivalence). *Two model parameter vectors $\theta_1 = (\beta_1, \Sigma_1)$ and $\theta_2 = (\beta_2, \Sigma_2)$ are **observationally equivalent** with respect to a set of shift distributions $\{\mathbb{P}^A_e : e \in \mathcal{E}_{\text{train}}\}$ [3] if they induce the same set $\mathcal{P}_{\theta,\mathcal{E}_{\text{train}}}$ of training distributions over the observed variables $(X_e, Y_e)$ as described in Section 2.1, i.e.*

$$\mathbb{P}^{X,Y}_{\theta_1,e} \cong \mathbb{P}^{X,Y}_{\theta_2,e} \text{ for all } e \in \mathcal{E}_{\text{train}}.$$

*By observing $\mathcal{P}_{\theta^\star,\mathcal{E}_{\text{train}}}$, we can identify the model parameters $\theta^\star$ up to the **observationally equivalent set** defined as*

$$\Theta_{\text{eq}} := \{\theta = (\beta, \Sigma) \in \Theta : \mathcal{P}_{\theta,\mathcal{E}_{\text{train}}} \cong \mathcal{P}_{\theta^\star,\mathcal{E}_{\text{train}}}\}.$$

In general, observationally equivalent set is not a singleton, that is, $\theta^\star$ is not identifiable from the collection of training environments $\mathcal{P}_{\theta^\star,\mathcal{E}_{\text{train}}}$. However, prior work has exclusively considered test shifts $M_{\text{test}}$ that still allow identifiability of the robust risk nonetheless, depicted in Figure 2a and discussed again in Section 3.2. In this work we argue for analyzing the more general partially identifiable setting, where set-identifiability of the invariant parameter $\theta^\star$ only allows us to compute a superset of the robustness set

$$\mathcal{P}_{\Theta_{\text{eq}}}(M_{\text{test}}) := \bigcup_{\theta \in \Theta_{\text{eq}}} \mathcal{P}_\theta(M_{\text{test}}) \supset \mathcal{P}_{\theta^\star}(M_{\text{test}})$$

and correspondingly, a set of robust risks $\{\mathcal{R}_{\text{rob}}(\beta; \theta, M_{\text{test}}) : \theta \in \Theta_{\text{eq}}\}$. In this case, we would still like to achieve the "best-possible" robustness, that is the test shift robustness for the "hardest-possible" parameters that could have induced the observed training distributions.

---

[3] The distributions $\mathbb{P}^A_e$ are to be understood up to the equivalence relation $\cong$. In general, the distributions $\mathbb{P}^A_e$ are unknown, since the shift variables $A_e$ are unobserved. However, in our setting, $\mathbb{P}^A_e$ can be identified up to the second moment because of the reference environment.

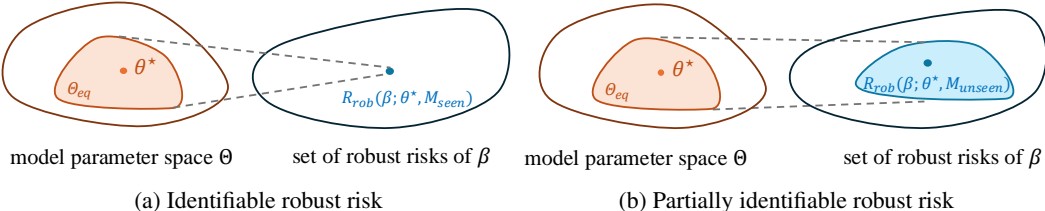

(a) Identifiable robust risk          (b) Partially identifiable robust risk

Figure 2: Relationship between identifiability of the model parameters and identifiability of the robust risk. (a) The classical scenario where the test shift upper bound $M_{\text{test}} = M_{\text{seen}}$ is contained in the span of training shifts so that the robust risk is point-identified. (b) The more general scenario of this paper, where $M_{\text{test}} = M_{\text{unseen}}$ contains new shift directions and where only a set can be identified in which the robust risk lies.

**Definition 2** (Worst-case robust risk and the minimax quantity)**.** *For the data model in Equation* (3), *the worst-case robust risk is defined as*

$$\mathfrak{R}_{\text{rob}}(\beta; \Theta_{\text{eq}}, M_{\text{test}}) := \sup_{\theta \in \Theta_{\text{eq}}} \mathcal{R}_{\text{rob}}(\beta; \theta, M_{\text{test}}). \tag{9}$$

*The optimal robustness on test shifts bounded by $M_{\text{test}}$ given training data $\mathcal{P}_{\theta^\star, \mathcal{E}_{\text{train}}}$ is described by the minimax quantity*

$$\mathfrak{M}(\Theta_{\text{eq}}, M_{\text{test}}) = \inf_{\beta \in \mathbb{R}^d} \mathfrak{R}_{\text{rob}}(\beta; \Theta_{\text{eq}}, M_{\text{test}}). \tag{10}$$

*When the minimizer of Equation* (10) *exists, we call it the worst-case robust predictor defined by*

$$\beta_{\Theta_{\text{eq}}}^{\text{rob}} = \arg\min_{\beta} \mathfrak{R}_{\text{rob}}(\beta; \Theta_{\text{eq}}, M_{\text{test}}) \tag{11}$$

In the next sections, we explicitly compute these quantities for the linear setting of Section 2. This will allow us to compare the best achievable robustness in the partially identified case with the guarantees of prior methods in this setting.

## 3 Theoretical results for the linear setting

We now compute the worst-case robust risk (9) and derive a lower bound for the minimax quantity (10) in the linear additive shift setting of Section 2. We then compare the worst-case robust risk of some existing robustness methods and ordinary least squares (OLS) with the minimizer of the worst-case robust risk both theoretically and empirically.

### 3.1 Minimax robustness results for the linear setting

The degree to which the model parameters $\theta^\star$ in the linear additive shift setting (3) can be identified depends on the number of environments and the total rank of the moments of the additive shifts. For structural causal models, this is well-studied, for instance, in the instrumental variable (IV) regression literature [4, 9]. As we show in Proposition 1, the true parameter $\beta^\star$ can *only* be identified along the directions of the training-time mean and variance shifts $\mu_e$ and $\Sigma_e$. Therefore, if not enough shift directions are observed, $\beta^\star$ is merely *set-identifiable*, leading to set-identifiability of the robust prediction model (8). More formally, we denote by $\mathcal{S}$ the subspace consisting of all *additive shift directions seen during training*:

$$\mathcal{S} := \text{range}\left[\sum_{e \in \mathcal{E}_{\text{train}}} \left(\Sigma_e + \mu_e \mu_e^\top\right)\right], \tag{12}$$

and by $\mathcal{S}^\perp$ its orthogonal complement. The definition of the space $\mathcal{S}$ induces an orthogonal decomposition of the true parameter $\beta^\star = \beta^{\mathcal{S}} + \beta^{\mathcal{S}^\perp}$. The *identifiable part* $\beta^{\mathcal{S}}$ then uniquely defines a set of *identified model parameters* that reads

$$\theta^{\mathcal{S}} := (\beta^{\mathcal{S}}, \Sigma_\eta^{\mathcal{S}}, \Sigma_{\eta,\xi}^{\mathcal{S}}, (\sigma_\xi^{\mathcal{S}})^2) = (\beta^{\mathcal{S}}, \Sigma_\eta^\star, \Sigma_{\eta,\xi}^\star + \Sigma_\eta^\star \beta^{\mathcal{S}}, (\sigma_\xi^\star)^2 + 2\langle \Sigma_{\eta,\xi}^\star, \beta^{\mathcal{S}} \rangle + \langle \beta^{\mathcal{S}}, \Sigma_\eta^\star \beta^{\mathcal{S}} \rangle)$$

that can be computed from the training distributions. For the following results, we assume a similar decomposition of the test shift upper bound $M_{\text{test}}$ which is essentially a decomposition into "seen" and "unseen" directions.

**Assumption 3.1** (Structure of $M_{\text{test}}$). *We assume that $M_{\text{test}} = \gamma M_{\text{seen}} + \gamma' R R^\top$, where $\gamma, \gamma' \geq 0$, $M_{\text{seen}}$ is a PSD matrix satisfying* range $M_{\text{seen}} \subset \mathcal{S}$ *and $R$ is a semi-orthogonal matrix satisfying* range $R \subset \mathcal{S}^\perp$.

In the next proposition, we show that the model parameters and robust predictor can be identified up to a neighborhood around $\theta^\mathcal{S}$.

**Proposition 1** (Identifiability of model parameters and robust predictor). *Suppose that the set of training and test distributions is generated according to Section 2.1 and Assumption 3.1 holds. Then,*

*(a) the model parameters generating the training distribution (3) can be identified up to the following observationally equivalent set:*

$$\Theta_{\text{eq}} = \Theta \cap \{\beta^\mathcal{S} + \alpha, \Sigma_\eta^\star, \Sigma_{\eta,\xi}^\mathcal{S} - \Sigma_\eta^\star \alpha, (\sigma_\xi^\mathcal{S})^2 - 2\alpha^\top \Sigma_{\eta,\xi}^\mathcal{S} + \alpha^\top \Sigma_\eta \alpha \colon \alpha \in \mathcal{S}^\perp\} \ni \theta^\star; \quad (13)$$

*(b) the robust predictor $\beta_\theta^{rob}$ as defined in Equation (8) is identified up to the set*

$$\mathcal{B}_{\Theta_{\text{eq}}}^{rob} \cap \{\beta^\mathcal{S} + (M_{\text{test}} + \Sigma_\eta^\star)^{-1} \Sigma_{\eta,\xi}^\mathcal{S} + (M_{\text{test}} + \Sigma_\eta^\star)^{-1} \alpha \colon \alpha \in \text{range } R\} \ni \beta_\theta^{rob}, \quad (14)$$

*where $\mathcal{B}_{\Theta_{\text{eq}}}^{rob} = \{\beta_\theta^{rob} : \theta \in \Theta_{\text{eq}}\}$.*

The proof of Proposition 1 is provided in Appendix F.1. Proposition 1 implies two well-known settings: If we observe a rich enough set $\mathcal{P}_{\mathcal{E}_{\text{train}}}$ of training environments such that $\mathcal{S} = \mathbb{R}^d$, the model parameters are uniquely identified, corresponding to the setting of full-rank instruments [4]. From a dual perspective, for a given set of training environments, *the robust predictor is identifiable whenever the test shifts are in the same direction as the training shifts*, i.e. range $M_{\text{test}} \subset \mathcal{S}$ and $R = 0$ – this holds even when the invariant parameters are not identifiable and $\mathcal{S} \neq \mathbb{R}^d$. This is the setting considered e.g. in anchor regression [41] and discussed again in Section 3.2 and Appendix C.

So far, we have described how the identifiability of the robust prediction model depends on the structure of both the training environments (via the space $\mathcal{S}$) and the test environments (via $M_{\text{test}}$). We now aim to compute the smallest achievable robust loss for the general partially identifiable setting, which allows for $R \neq 0$. In particular, we provide a lower bound on the *best-possible achievable distributional robustness* formalized by the minimax quantity (10). First observe that without further assumptions on the parameter space $\Theta$, the observationally equivalent set is unbounded, and the worst-case robust risk (9) can be infinite. The following boundedness assumption allows us to provide a fine-grained analysis of robustness in a partially identified setting.

**Assumption 3.2** (Boundedness of the causal parameter). *There exists a constant $C > 0$ such that any parameter $\beta$ in the DGP (3) is norm-bounded by $C$, i.e. $\|\beta\|_2 \leq C$.*

Furthermore, we define $C_{\text{ker}} = \sqrt{C^2 - \|\beta^\mathcal{S}\|^2}$, the maximum norm of the non-identified part of the linear parameter $\beta^\star$. Finally, recall that the reference distribution $\mathbb{P}_{\theta^\star,0}^{X,Y}$ is observed and hence identifiable.

**Theorem 3.1.** *Assume that the training and test data follow the data-generating mechanism in Section 2.1 and $M_{\text{test}}$ satisfies Assumption 3.1 for some $M_{\text{seen}}, R$ with* range $M_{\text{seen}} \subset \mathcal{S}$, range $R \subset \mathcal{S}^\perp$. *Further, let Assumption 3.2 hold with parameter $C$. The worst-case robust risk (9) is then given by*

$$\mathfrak{R}_{\text{rob}}(\beta; \Theta_{\text{eq}}, M_{\text{test}}) = \gamma'(C_{\text{ker}} + \|R^\top \beta\|_2)^2 + \gamma(\beta^\mathcal{S} - \beta)^\top M_{\text{seen}}(\beta^\mathcal{S} - \beta) + \mathcal{R}(\beta; \mathbb{P}_{\theta^\star,0}^{X,Y}), \quad (15)$$

*The minimax quantity in Equation (10) is lower bounded as follows:*

$$\mathfrak{M}(\Theta_{\text{eq}}, M_{\text{test}}) \begin{cases} = \gamma' C_{\text{ker}}^2 + \min_{R^\top \beta = 0} \mathcal{R}_{\text{rob}}(\beta; \theta^\mathcal{S}, \gamma M_{\text{seen}}), & \text{if } \gamma' \geq \gamma'_{\text{th}}; \\ \geq \gamma' C_{\text{ker}}^2 + \min_{\beta \in \mathbb{R}^d} \mathcal{R}_{\text{rob}}(\beta; \theta^\mathcal{S}, \gamma M_{\text{seen}}), & \text{else,} \end{cases} \quad (16)$$

*where $\gamma'_{\text{th}} = \frac{(\kappa(\Sigma_\eta^\star)+1)\|RR^\top \Sigma_{\eta,\xi}^\mathcal{S}\|}{C_{\text{ker}}}$ [4]. Moreover, for small unseen shifts*

$$\lim_{\gamma' \to 0} \frac{\mathfrak{M}(\Theta_{\text{eq}}, M_{\text{test}})}{\gamma'} = (C_{\text{ker}} + \|RR^\top \Sigma_\eta^{\star-1} \Sigma_{\eta,\xi}^\mathcal{S}\|)^2. \quad (17)$$

---

[4]$\kappa$ denotes the conditional number of the covariance matrix $\Sigma_\eta^\star$.

We prove Theorem 3.1 in Appendix F.2. First, in the case of no new test shifts where $\gamma' = 0$ (as it appears in prior work [41, 45]) we can plug in the robust risk Equation (7) into Equation (16) to observe the following: as the strength $\gamma$ of the shift grows, the optimal robust risk saturates. On the other hand, if $\gamma' \neq 0$, i.e., the test shift contains new directions w.r.t. to the training data, the best achievable robustness $\mathfrak{M}(\Theta_{\text{eq}}, M_{\text{test}})$ grows linearly with $\gamma'$. Further note that for $\gamma' \geq \gamma'_{\text{th}}$, we have a tight expression for the minimax quantity and the worst-case robust predictor $\beta^{\text{rob}}_{\Theta_{\text{eq}}}$ can be explicitly computed (cf. Appendix F.2) and is *orthogonal* to the space range $R$ of non-identifiable test shift directions. In other words, for large shifts in non-identified directions, the optimal robust model would "abstain" from prediction in those directions. For smaller $\gamma'$, $\beta^{\text{rob}}_{\Theta_{\text{eq}}}$ gradually utilizes less information in the non-identified directions, thus interpolating between maximum predictive power (OLS) and robustness w.r.t. new directions (abstaining). The model $\beta^{\text{rob}}_{\Theta_{\text{eq}}}$ is a population quantity that is identifiable from the collection of training *distributions*. When only finite samples are available, we discuss in Appendix D how we can compute the worst-case robust estimator by minimizing an empirical loss function that can be computed from multi-environment data.

### 3.2 Theoretical analysis of existing finite robustness methods

We now evaluate existing finite robustness methods in our partial identifiability framework and characterize their (sub)optimality in different scenarios. A spiritually similar systematic comparison of domain adaptation methods is presented in [12], however, in our setting, the robust risk is not identifiable from data. Concretely, we compare discrete anchor regression [41] and pooled OLS estimators [5] with the minimax quantity in Theorem 3.1. We consider the same scenario as in discrete anchor regression, which is a the specific case of the setting in Equation (3), where for each environment $e$, $A_e$ is just a mean shift with variance 0. In addition, discrete anchor regression assumes that the environment variable $E \in \mathcal{E}_{\text{train}}$ follows a probability distribution with $\mathbb{P}[E = e] = w_e$. The discrete anchor setting then corresponds to setting a test shift upper bound $M_{\text{test}} = \gamma M_{\text{anchor}}$ for some $\gamma > 0$ (cf. Equation (4)) with $M_{\text{anchor}} = \sum_{e \in \mathcal{E}_{\text{train}}} w_e \mu_e \mu_e^\top$. The (oracle) discrete anchor regression estimator minimizes the robust risk and reads

$$\beta_{\text{anchor}} = \arg\min_{\beta \in \mathbb{R}^d} \mathcal{R}_{\text{rob}}(\beta; \theta^\star, \gamma M_{\text{anchor}}), \tag{18}$$

The pooled ordinary least squares (OLS) estimator $\beta_{\text{OLS}}$ corresponds to $\beta_{\text{anchor}}$ with $\gamma = 1$. We observe that the test shifts bounded by $\gamma M_{\text{anchor}}$ are fully contained in the space of identified directions $\mathcal{S}$, since $\mathcal{S} = \text{range} \cup_{e \in \mathcal{E}_{\text{train}}} \mu_e \mu_e^\top = \text{range } M_{\text{anchor}}$. Thus, according to Proposition 1, the robust risk and robust predictor $\beta_{\text{anchor}}$ are identifiable for all $\gamma > 0$. In the next corollary, we compute worst-case robust risk of both $\beta_{\text{anchor}}$ and $\beta_{\text{OLS}}$ with respect to the more general shifts bounded by $M_{\text{test}} := \gamma M_{\text{anchor}} + \gamma' R R^\top$, thus possibly including unseen shifts consisting of additional unseen shifts range $R \subset \mathcal{S}^\perp$.

**Corollary 3.2** (Worst-case robust risk of the anchor regression estimator). *Assume that the test shift upper bound is given by $M_{\text{test}} := \gamma M_{\text{anchor}} + \gamma' R R^\top$. Let $\mathbb{P}^{X,Y}_{\text{train}} = \sum_e w_e \mathbb{P}^{X,Y}_e$ be the pooled training distribution. Then the general worst-case robust risk is given by*

$$\mathfrak{R}_{\text{rob}}(\beta; \Theta_{\text{eq}}, M_{\text{test}}) = \gamma'(C_{\text{ker}} + \|R^\top \beta\|_2)^2 + (\gamma - 1)(\beta^\mathcal{S} - \beta)^\top M_{\text{anchor}}(\beta - \beta^\mathcal{S}) + \mathcal{R}(\beta, \mathbb{P}^{X,Y}_{\text{train}}).$$

*Furthermore, the the anchor and OLS predictor, respectively, it holds that there exists constants $c_1, c_2, c_3$ independent of $\gamma, \gamma'$ such that*

$$\mathfrak{R}_{\text{rob}}(\beta_{\text{anchor}}; \Theta_{\text{eq}}, M_{\text{test}}) = (C_{\text{ker}} + \|R R^\top (\Sigma^\star_\eta + \gamma M_{\text{anchor}})^{-1} \Sigma^\mathcal{S}_{\eta,\xi}\|)^2 \gamma' + c_1;$$

$$\mathfrak{R}_{\text{rob}}(\beta_{\text{OLS}}; \Theta_{\text{eq}}, M_{\text{test}}) = (C_{\text{ker}} + \|R R^\top (\Sigma^\star_\eta + M_{\text{anchor}})^{-1} \Sigma^\mathcal{S}_{\eta,\xi}\|)^2 \gamma' + c_2.$$

*In contrast, the best achievable robustness reads*

$$\mathfrak{M}(\Theta_{\text{eq}}, M_{\text{test}}) = C^2_{\text{ker}} \gamma' + c_3, \quad \text{if } \gamma' \geq \gamma'_{\text{th}};$$

$$\lim_{\gamma' \to 0} \mathfrak{M}(\Theta_{\text{eq}}, M_{\text{test}})/\gamma' = (C_{\text{ker}} + \|R R^\top (\Sigma^\star_\eta + \gamma M_{\text{anchor}})^{-1} \Sigma^\mathcal{S}_{\eta,\xi}\|)^2.$$

---

[5] In Appendix C we show that analogous results hold for continuous anchor regression and the method of distributionally robust invariant gradients (DRIG) [45].

Observe that the worst-case robust risk in the extended anchor regression setting is equal to the anchor regression risk with an additional non-identifiability penalty term $\gamma'(C_{\text{ker}} + \|R^\top \beta\|_2)^2$. The anchor regression estimator is optimal in the limit of vanishing unseen shifts but (for any $\gamma$) significantly deviates[6] from the best achievable robustness for larger unseen shifts $\gamma' \geq \gamma'_{\text{th}}$. Moreover, in case of completely new shifts ($\gamma = 0$), pooled OLS and the anchor estimator achieve the same rate in $\gamma'$, showcasing how finite robustness methods can perform similarly to empirical risk minimization if the assumptions on the robustness set are not met. We provide additional performance comparisons for the more general shift in Appendix C and the proof of the corollary in Appendix F.3.

## 4 Experimental results

In this section, we provide empirical evidence of our theoretical conclusions in Sections 3.1 and 3.2. In particular, we compare the prediction performance of multiple existing robustness methods to the (estimated) minimax robustness in identifiable and partially identifiable settings. We observe that both on synthetic and real-world data, in the partially identified setting, empirical risk minimization and invariance-based robustness methods not only have significantly sub-optimal test loss, but also perform more similarly, thereby aligning with our theoretical results in Section 3.2. This stands in contrast to the identifiable setting, where the anchor predictor is optimal up to finite-sample effects. Furthermore, we observe that even though the minimizer of the worst-case robust risk is optimal only for the linear causal setting in Section 2.1, it surprisingly outperforms existing methods in a real-world experiment.

**Experiments on synthetic Gaussian data** We simulate Gaussian covariates according to Equation (3) with multiple environments differing by linearly independent randomly selected mean shifts. For a randomly sampled collection of mean shifts, we evaluate a proxy for the worst-case robust risk by picking the most adversarial $(\beta^\star, \Sigma_\eta^\star)$ for the shifts, and then computing its robust risk (7). We describe the full details of the data generation and loss evaluation in Appendix E.1. We consider two shift scenarios: in the identifiable case in (see Figure 2a), the test environment is only perturbed by bounded shifts in training directions with increasing strength $\gamma$, as considered in prior work [41, 45]. In the non-identifiable case (see Figure 2b), the test environment is perturbed by a mixture of training shifts and shifts in previously unobserved directions, where $\gamma$ is fixed and $\gamma'$ varies (cf. Assumption 3.1). We compute the empirical minimizers $\hat{\beta}_{\text{OLS}}, \hat{\beta}_{\text{anchor}}$ and $\hat{\beta}_{\Theta_{\text{eq}}}^{\text{rob}}$ of the OLS, anchor regression and worst-case robust losses, respectively, and compare their worst-case robust risk (mean squared error) in Figure 3. In the identifiable setting – Figure 3 (left) – the robust risk is asymptotically constant across $\gamma$ for both robust methods, while the error for the OLS, or vanilla ERM, estimator increases linearly. In contrast, in the second, partially identified, setting – Figure 3 (right) – all estimators exhibit linearly increasing test errors; however the slopes of the anchor and OLS estimator are much steeper and lead to larger errors than the empirical minimizer of (15) that closely matches the analytic theoretical lower bound.

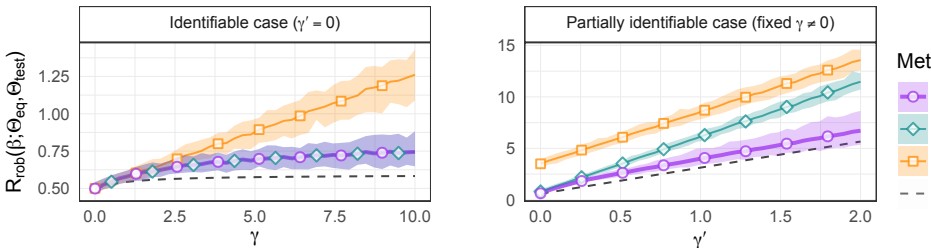

Figure 3: Worst-case robust risk of the baseline estimators $\beta_{\text{OLS}}, \beta_{\text{anchor}}$ (using the "correct" $\gamma$), the worst-case robust predictor in (mean-shifted) multi-environment finite-sample experiments and theoretical population lower bound in the classical identified setting with varying shift strength $\gamma$ (left) and the partially identifiable setting with fixed $\gamma$ but varying $\gamma'$ (right). The details of the experimental setting can be found in Appendix E.

**Real-world data experiments** We evaluate the performance of OOD methods using single-cell gene expression data from [38], consisting of $d = 622$ genes across observational and interventional environments. As in [44], we focus on 28 genes active in the observational environment. For each gene $j = 1, \ldots, 28$, we define the target variable $Y := X_j$ and select the three genes most strongly

---
[6]Notice that the term $\|RR^\top(\Sigma_\eta^\star + \gamma M_{\text{anchor}})^{-1}\Sigma_{\eta,\xi}^{\mathcal{S}}\|$ generally only goes to zero as $\gamma \to \infty$ (yielding the minimax risk) if $M_{\text{anchor}}$ is full-rank, otherwise, it can be strictly bounded from below as $\Sigma_\eta^\star$ is full-rank.

correlated with $Y$ as covariates. This yields 28 prediction problems indexed by $j$, each consisting of data from an observational environment $\mathcal{O}$ and three interventional environments $\mathcal{I}_{j1}, \mathcal{I}_{j2}, \mathcal{I}_{j3}$ representing the gene knockout on a single covariate. For each prediction problem, we consider three training datasets $D_{j1}, D_{j2}, D_{j3}$, obtained by combining data from $\mathcal{O}$ with a single interventional environment $\mathcal{I}_{j1}, \mathcal{I}_{j2}, \mathcal{I}_{j3}$, respectively. For each training dataset $D_{jk}$, $k = 1, 2, 3$, we evaluate the mean-squared error (MSE) at test time using four datasets consisting of varying proportions of unseen shifts (e.g., "33% unseen directions" in Figure 4 represents a test dataset with 67% observations sampled from $\mathcal{I}_{jk}$ and 33% from $\mathcal{I}_{j\ell}$ with $\ell \neq k$). Hence, for each prediction problem predicting a gene $j$, we evaluate on 12 configurations (three training and four test datasets).[7] Figure 4 illustrates the test MSE of the worst-case robust estimator (Worst-case Rob.) alongside anchor regression, invariant causal prediction (ICP), DRIG, and OLS, as a function of perturbation strength $s$. [8] For a given proportion of unseen shifts, $s$ controls the distance of the test data points from the observational mean, acting as a proxy for shift strengths $\gamma$ and $\gamma'$. [9] We observe that the performance ranking of the robustness methods significantly varies with the proportion of new test shift directions. As expected, when no new shift directions are present at test time (0%), anchor regression and DRIG are optimal, since they protect against shifts observed at training time. However, as soon as some unseen directions are present, their performance becomes inferior to OLS/ERM and the gap to the worst-case robust predictor (in the linear setting described in Section 2) grows with the proportion of unseen shifts. Further, while the MSE of the previous invariant methods increases significantly with the strength of the test shift $s$, the test loss of the worst-case robust predictor remains relatively stable.

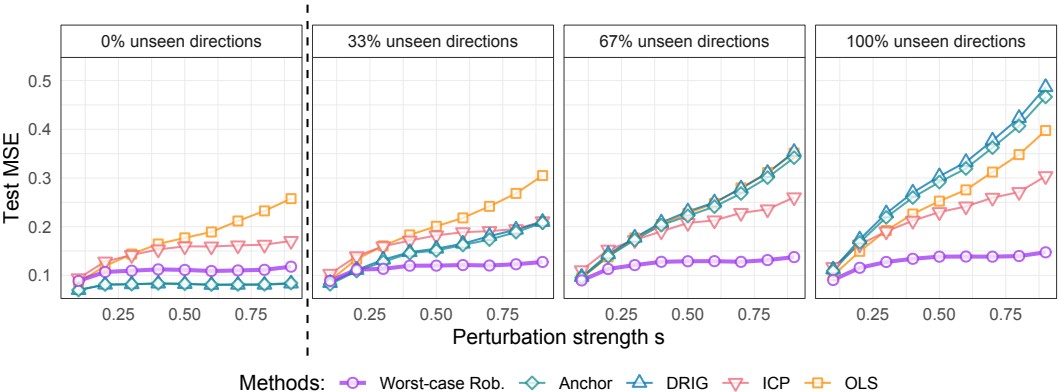

Figure 4: The figures show the performance of the *worst-case robust predictor* (Worst-case Rob.) compared to other methods as a function of perturbation strength $s$. Different panels correspond to the proportion of unseen shift directions at test time. For each panel and perturbation strength $s$, each point represents an average over the 28 target genes and three experiments (i.e., training environments).

## 5    Conclusion and future directions

This paper introduces the worst-case robust risk – a quantity that is well-defined even in settings where the usual robust risk is not computable from training distributions, and in identifiable scenarios [41, 45] reduces to the conventional robust risk. We instantiate our general framework for linear models with additive distribution shifts and compute tight lower bounds for this setting. Further, we demonstrate how i) the benefits of invariance-based robustness methods strongly decrease in the partially identifiable setting; and ii) this suboptimality increases with perturbation strength and proportion of previously unobserved test shifts.

The main limitation of our paper is its reliance on a linear setting to explicitly compute the worst-case robust risk and estimate the minimax quantity. However, we expect that the results and intuition developed in this paper can be extended to linear shifts in a lower-dimensional latent space via a suitable parametric or non-linear map [51, 10]. Important future directions include extending our results to more general shift models, non-linear functional relationships and the classification setting. Further, a potential use of our work is in the field of *active intervention selection* (e.g, [57, 20]). By computing the most adversarial model parameter for a given estimator, e.g., OLS, we can obtain an intervention which minimizes the worst-case robust risk of the estimator on the next unseen shift.

---

[7]An illustration of the training and test setups can be found in Figure 5.

[8]Details on the tuning parameter for each method are in Appendix E.2.

[9]More details on the shift strength can be found in Appendix E.2.

# 6 Acknowledgements

JK was supported by the SNF grant number 204439. NG was supported by the SNF grant number 210976. We thank Kasra Jalaldoust and Yixin Wang for helpful discussions and feedback on the manuscript.

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

# Appendix

The following sections provide deferred discussions, proofs and experimental details.

**Table of contents**

# A  Extended related work

To put our work into context, first, we discuss relevant distributional robustness literature organized according to structural assumptions on the desired robustness set. Second, we summarize existing views on partial identifiability in the causality and econometrics literature and how our findings connect to their perspective.

*No structural assumptions on the shift.* **DRO:** Distributionally robust optimization (DRO) tackles the problem of domain generalization when the robustness set is a ball around the training distribution w.r.t. some probability distance measure, e.g., Wasserstein distance [49, 32] or $f$-divergences [7, 15]. Considering all test distributions in a discrepancy ball can lead to overly conservative predictions, and therefore, alternatives have been proposed in, e.g., the Group DRO literature [43, 19, 29]. However, these methods cannot protect against perturbations larger than the ones seen during training time and do not provide a clear interpretation of the perturbations class [45].

*Structural assumptions on the shift.* Robustness from the lens of causality takes a step further, by assuming a structural causal model [34] generating the observed data $(X, Y)$. **Infinite robustness methods:** The motivation of causal methods for robustness is that the causal function is worst-case optimal to predict the response under interventions of arbitrary direction and strength on the covariates [31, 11]. For this reason, causal models achieve what we call *infinite robustness*. Depending on the assumptions of the SCM, there are different ways to achieve infinite robustness. When there are no latent confounders, several works [35, 17, 30, 39, 2, 6, 46, 54, 28, 1] aim to identify the causal parents and achieve infinite robustness by exploiting the heterogeneity across training environments. In the presence of latent confounders, it is possible to achieve infinite robustness by identifying the causal function with, e.g., the instrumental variable method [5, 22, 48, 8, 33]. There are different limitations to *infinitely robust* methods. First, the identifiability conditions of the causal parents and/or causal function are often challenging to verify in practice. Second, ERM can outperform these methods when the interventions (read shifts) at test time are not arbitrarily strong or act directly on the response or latent variable [3, 21]. **Finite robustness methods:** In real data, shifts of arbitrary direction and strength in the covariates are unrealistic. Thus, different methods [41, 23, 27, 45, 14] trade off robustness against predictive power to achieve what we call *finite robustness*. The main idea of finite robustness methods is to learn a function that is as predictive as possible while protecting against shifts up to some strength in the directions that are observed during training time. These methods, however, only provide robustness guarantees that depend on the heterogeneity of the training data and do not offer insights into the limits of *algorithm-independent robustness* under shifts in new directions.

*Partial identifiability*: The problem of identification is at the center of the causal and econometric literature [36, 4]. It studies the conditions under which the (population) training distribution uniquely determines the causal parameters of the underlying SCM. Often, the training distribution only offers partial information about the causal parameters and, therefore, determines a set of observational equivalent parameters. This setting is known as *partial* or *set identification* and is used in causality and econometrics to learn intervals within which the true causal parameter lies [50]. In this work, we borrow the notion of partial identification to study the problem of distributional robustness when the robustness set itself is only partially identified.

# B  Extension to the general additive shift setting

We discuss how our setting changes when we relax the assumptions on the existence of the reference environment. We consider the data-generating process in Equation (3), where $\mathcal{E}_{\text{train}} = [m]$, $m \in \mathbb{N}$. If no environment $e$ exists with $\mu_e = 0$ and $\Sigma_e = 0$, we first pick an arbitrary distribution $\mathbb{P}_{\text{ref}}^{X,Y}$ as the reference environment[10] . We denote $\Sigma'_\eta := \Sigma_\eta^\star + \Sigma_{\text{ref}}$.

First, we show we can express the space $\mathcal{S}$ of training additive shift directions defined in Equation (12) in the general case. We center all distributions by $\mu_{\text{ref}}$ to obtain centered distributions $\tilde{\mathbb{P}}\text{r}$ that

---

[10]In practice, it is useful to pick a distribution with the smallest covariance, i.e. $\operatorname{tr}\operatorname{Cov}(X_{\text{ref}}) \leq \operatorname{tr}\operatorname{Cov}(X_e)$ for all $e$.

$\mathbb{E}_{X \sim \tilde{P}_e}[X] = 0$. With respect to the arbitrary reference environment, we now define
$$\tilde{\mathcal{S}} := \text{range} \bigcup_{e \in \mathcal{E}_{\text{train}}} \left(\Sigma_e - \Sigma_{\text{ref}} + (\mu_e - \mu_{\text{ref}})(\mu_e - \mu_{\text{ref}})^\top\right) \subset \mathbb{R}^d.$$

We now consider test shifts with respect to the environment $\mathbb{P}_{\text{ref}}^{X,Y}$[11]. We define the test shift upper bound $M_{\text{test}} = \gamma M_{\text{seen}} + \gamma' RR^\top$, where range $M_{\text{seen}} \subset \mathcal{S}$ and range $R \subset \mathcal{S}^\perp$. Again, we can decompose the parameter $\beta^\star$ as $\beta^\star = \beta^\mathcal{S} + \beta^{\mathcal{S}^\perp}$. The projection $\beta^\mathcal{S}$ of the causal parameter onto the relative training shifts induces the following observationally equivalent parameters corresponding to the reference distribution:
$$\theta^\mathcal{S} := (\beta^\mathcal{S}, \Sigma'_\eta, \Sigma_{\eta,\xi}^\mathcal{S}, (\sigma_\xi^\mathcal{S})^2) = (\beta^\mathcal{S}, \Sigma'_\eta, \Sigma_{\eta,\xi}^\star + \Sigma'_\eta \beta^{\mathcal{S}^\perp}, (\sigma_\xi^\star)^2 + 2\langle \Sigma_{\eta,\xi}^\star, \beta^{\mathcal{S}^\perp}\rangle + \langle \beta^{\mathcal{S}^\perp}, \Sigma'_\eta \beta^{\mathcal{S}^\perp}\rangle).$$

Again, $\theta^\mathcal{S}$ can be identified from the training distributions and is referred to as the *identified model parameters*. The following adapted version of Proposition 1 shows that assuming shifts on $\mathbb{P}_{\text{ref}}^{X,Y}$, the robust prediction model is only identifiable if the test shifts are in the direction of the relative training shifts:

**Proposition 2** (Identifiability of reference distribution parameters and robust prediction model)**.** *Suppose that the set of training and test distributions is generated according to Equations* (3) *and* (4)*. Then, $\theta^\mathcal{S}$ is observationally equivalent to $\theta^\star$ and computable from training distributions. Furthermore, it holds that*

(a) *the model parameters generating the reference distribution can be identified up to the following observationally equivalent set :*
$$\Theta_{\text{eq}} = \{\beta^\mathcal{S} + \alpha, \Sigma'_\eta, \Sigma_{\eta,\xi}^\mathcal{S} - \Sigma'_\eta \alpha, (\sigma_\xi^\mathcal{S})^2 - 2\alpha^\top \Sigma_{\eta,\xi}^\mathcal{S} + \alpha^\top \Sigma'_\eta \alpha : \alpha \in \mathcal{S}^\perp\} \ni \theta^\star$$

(b) *the robust prediction model $\beta^{rob}$ as defined in Equation* (8) *is identified up to the set*
$$\beta^\mathcal{S} + (\gamma\Pi_\mathcal{M} + \Sigma'_\eta)^{-1}\Sigma_{\eta,\xi}^\mathcal{S} + \{(\gamma\Pi_\mathcal{M} + \Sigma'_\eta)^{-1}\alpha : \alpha \in \text{range } R\} \ni \beta^{rob}$$

The proof is analogous to Appendix F.1. A version of Theorem 3.1 for perturbations on the reference environment follows accordingly.

## C Comparison to finite robustness methods continued

### C.1 The setting of continuous anchor regression [41]

In this section, we evaluate the worst-case robust risk of the continuous anchor regression estimator. In the continuous anchor regression setting, during training we observe the distribution according to the process $X = MA + \eta; Y = \beta^{\star\top}X + \xi$, where $A$ is an observed $q$-dimensional anchor variable with mean 0 and covariance $\Sigma_A$ and $M \in \mathbb{R}^{d \times q}$ is a known matrix. Note that in this setting, we do not have a reference environment, but, since the anchor variable is observed, the distribution of the additive shift $MA$ is known. The test shifts are assumed to be bounded by $M_{\text{test}} = \gamma M\Sigma_A M^\top$. Since range $M_{\text{test}} \subset \mathcal{S} = \text{range } M$, no new directions are observed during test time, in other words, $R = 0$. Thus, both the corresponding robust loss and the anchor regression estimator can be determined from training data. It holds that
$$\beta_{\text{anchor}} = \arg\min_{\beta \in \mathbb{R}^d} \mathcal{R}_{\text{rob}}(\beta; \theta^\star, \gamma M\Sigma_A M^\top).$$

Again, the pooled OLS estimator corresponds to $\beta_{\text{anchor}}$ with $\gamma = 1$. Similar to the discrete anchor case, in case the test shifts are given by $M_{\text{new}} = \gamma M\Sigma_A M^\top + \gamma' RR^\top$, the worst-case robust risk (9) is given by
$$\mathfrak{R}_{\text{rob}}(\beta; \Theta_{\text{eq}}, M_{\text{new}}) = \gamma'(C_{\text{ker}} + \|R^\top \beta\|_2)^2 + \mathcal{R}_{\text{rob}}(\beta; \theta^\star, \gamma M\Sigma_A M^\top)$$
and for the best worst-case robustness of the anchor estimator it holds
$$\mathfrak{R}_{\text{rob}}(\beta_{\text{anchor}}, \Theta_{\text{eq}}; M_{\text{test}}) = (C_{\text{ker}} + \|RR^\top(\Sigma_\eta^\star + \gamma M\Sigma_A M^\top)^{-1}\Sigma_{\eta,\xi}^\mathcal{S}\|)^2\gamma' + \text{const};$$
$$\lim_{\gamma' \to 0} \mathfrak{R}_{\text{rob}}(\beta_{\text{anchor}}, \Theta_{\text{eq}}; M_{\text{new}})/\gamma' = \lim_{\gamma' \to 0} \mathfrak{M}(\Theta_{\text{eq}}, M_{\text{new}})/\gamma'.$$

The above results follow by analogy with Appendix F.3.

---

[11] In other words, we require that the test distribution is a shifted version of the (arbitrarily) chosen reference distribution.

## C.2 Distributionally robust invariant gradients (DRIG) [45]

DRIG [45] introduce a more general additive shift framework, where a collection of additive shifts $A_e$ is given with moments $(\mu_e, \Sigma_e)$. For each environment $e$, we observe data $(X_e, Y_e)$ distributed according to the equations $X_e = A_e + \eta;\ Y_e = \beta^{\star\top} X_e + \xi$, where the noise is distributed like in Equation (3). This DGP arises from the structural causal model assumption as described in Figure 1. DRIG consider more a more general intervention setting, additionally allowing additive shifts of $Y$ and hidden confounders $H$. However, their identifiability results can only be shown for the case of interventions on $X$, and since identifiability of the causal parameter is a crucial part of our analysis, we only consider shifts on the covariates. DRIG assumes existence of a reference environment $e = 0$ with $\mu_0 = 0$ and for which it is required that the second moment of the reference environment is dominated by the second moment of the training mixture:

$$\Sigma_0 \preceq \sum_{e \in [m]} w_e (\Sigma_e + \mu_e \mu_e^\top).$$

This assumption allows [45] to derive the DRIG estimator which is robust against test shifts upper bounded by $M_{\mathrm{DRIG}} := \gamma \sum_{e \in [m]} w_e (\Sigma_e - \Sigma_0 + \mu_e \mu_e^\top)$. The following lemma allows us to make further statements about $M_{\mathrm{DRIG}}$:

**Lemma C.1.** *Let $A$ and $B$ be positive semidefinite matrices such that $B \preceq A$. Then it holds that range $B \subset$ range $A$.*

*Proof.* It suffices to show that $\ker A \subset \ker B$. ($\ker A \subset \ker B$ implies that range $A = (\ker A)^\perp \subset (\ker B)^\perp =$ range $B$.) Consider $x \in \ker A$, $x \neq 0$. Then it holds that $x^\top (A - B) x = x^\top A x - x^\top B x = 0 - x^\top B x \geq 0$, from which it follows that $x^\top B x = 0$ and thus $x \in \ker B$. $\qquad\square$

Because of the assumption $\Sigma_0 \preceq \sum_{e \in [m]} w_e (\Sigma_e + \mu_e \mu_e^\top)$, by Lemma C.1 it follows that range $\Sigma_0 \subset \cup_{e \geq 1}$ range $(\Sigma_e + \mu_e \mu_e^\top)$ and thus

$$\text{range } M_{\mathrm{DRIG}} \subseteq \text{range } \left( \sum_{e \geq 1} w_e (\Sigma_e + \mu_e \mu_e^\top) \right).$$

Hence, the robustness directions achievable by DRIG in the "dominated reference environment" setting are the same as the ones under the assumption $\Sigma_0 = 0$.

Again, we observe that the test shifts bounded by $\gamma M_{\mathrm{DRIG}}$ are fully contained in the space of identified directions $\mathcal{S}$. If the test shifts are instead bounded by $M_{\mathrm{new}} := \gamma M_{\mathrm{DRIG}} + \gamma' RR^\top$, including some unseen directions range $R \subset \mathcal{S}^\perp$, the robust risk in the DRIG setting is only partially identified. The worst-case robust risk (9) is given by

$$\mathfrak{R}_{\mathrm{rob}}(\beta; \Theta_{\mathrm{eq}}, M_{\mathrm{new}}) = \gamma'(C_{\mathrm{ker}} + \|R^\top \beta\|_2)^2 + \mathcal{R}_{\mathrm{rob}}(\beta; \theta^\star, \gamma M_{\mathrm{DRIG}}),$$

and again, the DRIG estimator is optimal for infinitesimal shifts $\gamma'$ and suboptimal for larger $\gamma'$:

$$\mathfrak{R}_{\mathrm{rob}}(\beta_{\mathrm{DRIG}}; \Theta_{\mathrm{eq}}, M_{\mathrm{new}}) = (C_{\mathrm{ker}} + \|RR^\top (\Sigma_\eta^\star + \gamma M_{\mathrm{DRIG}})^{-1} \Sigma_{\eta, \xi}^{\mathcal{S}}\|)^2 \gamma' + \text{const};$$

$$\text{whereas } \frac{\mathfrak{M}(\Theta_{\mathrm{eq}}, M_{\mathrm{new}})}{\gamma'} = C_{\mathrm{ker}}^2, \ \ \text{if } \gamma' \geq \gamma'_{\mathrm{th}};$$

$$\lim_{\gamma' \to 0} \frac{\mathfrak{M}(\Theta_{\mathrm{eq}}, M_{\mathrm{new}})}{\gamma'} = (C_{\mathrm{ker}} + \|RR^\top (\Sigma_\eta^\star + \gamma M_{\mathrm{DRIG}})^{-1} \Sigma_{\eta, \xi}^{\mathcal{S}}\|)^2.$$

The above results follow by plugging $M_{\mathrm{new}}$ with $M := M_{\mathrm{DRIG}}$ into the proof of Corollary 3.2 in Appendix F.3..

# D Empirical estimation of the worst-case robust predictor

In this section, we discuss how to compute the worst-case robust loss and its minimizer from finite-sample multi-environment training data. We first describe the finite-sample setting and provide a high-level algorithm. We then discuss some parts of the algorithm in more detail. Finally, we show that the empirical worst-case robust loss is consistent under certain assumptions. For simplicity, in this section, we assume that $M_{\mathrm{test}} = \gamma SS^\top + \gamma' RR^\top$, where $\gamma, \gamma' \geq 0$, $SS^\top$ is a semi-orthogonal matrix satisfying range $SS^\top \subset \mathcal{S}$ and $R$ is a semi-orthogonal matrix satisfying range $R \subset \mathcal{S}^\perp$. However, the results and strategies in this section can be easily applied to more general $M_{\mathrm{test}}$.

## D.1 Computing the worst-case robust loss

---

**Algorithm 1** Computation of the worst-case robust loss

---

1: **Input:** Multi-environment data $\mathcal{D} := \cup_{e \in \mathcal{E}_{\text{train}}} \mathcal{D}_e$, test shift strength $\gamma > 0$, test shift directions $M \in \mathbb{R}^{d \times d}$, causal parameter upper bound $C > 0$.
2: **Step 1:** Estimate the training shift directions $\hat{\mathcal{S}}(\mathcal{D})$, its orthogonal complement $\hat{\mathcal{S}}^\perp(\mathcal{D})$, and the identified causal parameter $\hat{\beta}^{\mathcal{S}}$.
3: **Step 2:** Estimate the identified and non-identified test shift directions $\hat{S}$, $\hat{R}$ and their projections $\hat{S}\hat{S}^\top$ and $\hat{R}\hat{R}^\top$.
4: **Step 3:** Estimate the norm $\hat{C}_{\text{ker}}$ of the non-identified causal parameter.
5: **Step 4:** Compute the worst-case robust loss function

$$\mathcal{L}_n(\beta; \hat{\beta}^{\mathcal{S}}, \hat{S}\hat{S}^\top, \hat{R}\hat{R}^\top) \leftarrow \underbrace{\mathcal{L}_{\text{ref}}(\beta; \mathcal{D}_0)}_{\text{reference loss}} + \underbrace{\mathcal{L}_{\text{inv}}(\beta; \hat{\beta}^{\mathcal{S}}, \hat{S}\hat{S}^\top, \gamma)}_{\text{invariance penalty term}} + \underbrace{\mathcal{L}_{\text{id}}(\beta; \hat{C}_{\text{ker}}, \hat{R}\hat{R}^\top, \gamma)}_{\text{non-identifiability penalty term}}.$$

6: **Return:** worst-case robust predictor and the estimated minimax "hardness" of the problem:

$$\hat{\beta}^{\text{rob}}_{\Theta_{\text{eq}}} \leftarrow \underset{\beta \in \mathbb{R}^d}{\arg \min} \, \mathcal{L}_n(\beta; \hat{\beta}^{\mathcal{S}}, \hat{S}\hat{S}^\top, \hat{R}\hat{R}^\top);$$

$$\hat{\mathfrak{M}}(\mathcal{D}, \gamma, M) \leftarrow \underset{\beta \in \mathbb{R}^d}{\min} \, \mathcal{L}_n(\beta; \hat{\beta}^{\mathcal{S}}, \hat{S}\hat{S}^\top, \hat{R}\hat{R}^\top).$$

---

**Training data.** We observe data from $m + 1$ training environments indexed by $E \in \mathcal{E}_{\text{train}} = \{0, ..., m\}$, where $E = 0$ represents the reference environment. We impose a discrete probability distribution $\mathbb{P}^E$ on the training environment $E \in \mathcal{E}_{\text{train}}$, resulting in the joint distribution $(X, Y, E) \sim \mathbb{P}^{X,Y|E} \times \mathbb{P}^E$. For each environment $E = e$, we observe the samples $\mathcal{D}_e := \{(X_{e,i}, Y_{e,i})\}_{i=1}^{n_e}$, where $(X_{e,i}, Y_{e,i})$ are independent copies of $(X_e, Y_e) \sim \mathbb{P}^{X,Y|E=e}$. Then, the resulting dataset is $\mathcal{D} := \cup_{e \in \mathcal{E}_{\text{train}}} \mathcal{D}_e$ with $n := n_0 + \cdots + n_m$. Furthermore, for each environment $E = e$, we define the weights $w_e := n_e/n$.

**Computation of the worst-case robust loss.** In Algorithm 1, we present a high-level scheme for computing the worst-case robust loss from multi-environment data, which consists of multiple steps. First, nuisance parameters related to the training and test shift directions are estimated, which we describe in more detail below. Afterwards, the three terms of the loss are computed: the (squared) loss $\mathcal{L}_{\text{ref}}(\beta; \mathcal{D}_0)$ on the reference environment is computed as

$$\mathcal{L}_{\text{ref}}(\beta; \mathcal{D}_0) = \sum_{i=1}^{n_0} (Y_{0,i} - \beta^\top X_{0,i})^2.$$

The invariance penalty term $\mathcal{L}_{\text{inv}}(\beta; \hat{\beta}^{\mathcal{S}}, \hat{S}\hat{S}^\top, \gamma)$ (which increasingly aligns any estimator $\beta$ in the direction of the estimated invariant causal predictor $\hat{\beta}^{\mathcal{S}}$ as $\gamma \to \infty$) can be computed as following in the linear SCM setting:

$$\mathcal{L}_{\text{inv}}(\beta; \hat{\beta}^{\mathcal{S}}, \hat{S}\hat{S}^\top, \gamma) = \gamma \|\hat{S}\hat{S}^\top(\beta - \beta^{\mathcal{S}})\|_2^2.$$

Finally, the non-identifiability penalty term $\mathcal{L}_{\text{id}}(\beta; \hat{C}_{\text{ker}}, \hat{R}\hat{R}^\top, \gamma)$ can be computed as follows:

$$\mathcal{L}_{\text{id}}(\beta; \hat{C}_{\text{ker}}, \hat{R}\hat{R}^\top, \gamma) \leftarrow \gamma(C_{\text{ker}} + \|\hat{R}\hat{R}^\top \beta\|_2)^2.$$

The non-identifiability term, with increasing $\gamma$, penalizes any predictor $\beta$ towards zero on the subspace $R$ of non-identified test shift directions. In total, the worst-case robust loss (in the linear SCM setting) equals

$$\mathcal{L}_n(\beta; \hat{\beta}^{\mathcal{S}}, \hat{S}\hat{S}^\top, \hat{R}\hat{R}^\top) = \sum_{i=1}^{n_0} (Y_{0,i} - \beta^\top X_{0,i})^2 + \gamma \|\hat{S}\hat{S}^\top(\beta - \beta^{\mathcal{S}})\|_2^2 + \gamma(C_{\text{ker}} + \|\hat{R}\hat{R}^\top \beta\|_2)^2,$$

where we suppress dependence on $C$ and $\gamma$ and only leave the dependence on the nuisance parameters.

**Choice/Estimation of nuisance parameters.** We now provide more details on the empirical estimation of the nuisance parameters $\hat{\mathcal{S}}, \hat{S}, \hat{R},$ and $\hat{\beta}^{\mathcal{S}}$.

- The **constant** $C$ corresponds to the upper bound on the norm of the true causal parameter $\beta^\star$. Thus, the practitioner chooses $C$ in advance to ensure that (with high probability) $\|\beta^\star\|_2 \le C$.

- The **training shift directions** $\hat{\mathcal{S}}$ can be computed via

$$\hat{\mathcal{S}}(\mathcal{D}) = \text{range} \sum_{e=1}^{m} (\text{Cov}(X^e) - \text{Cov}(X^0) + \mu_e \mu_e^\top - \mu_0 \mu_0^\top), \qquad (19)$$

  where for $e \in \mathcal{E}_{\text{train}}$, the matrix $\text{Cov}(X^e)$ is the empirical covariance matrix estimated within the training environment $E = e$, and $\mu_e \in \mathbb{R}^d$ is the empirical mean of the covariates within the training environment $E = e$. Additionally, we compute the orthogonal complement $\hat{\mathcal{S}}^\perp(\mathcal{D})$ of the space $\hat{\mathcal{S}}(\mathcal{D})$[12].

- The **decomposition of the test shift directions** $M$ into identified and non-identified shift directions (and their corresponding projection matrices) can be computed as follows. Let $\Pi_{\hat{\mathcal{S}}}$ and $\Pi_{\hat{\mathcal{S}}^\perp}$ denote the projection matrices on $\hat{\mathcal{S}}(\mathcal{D})$ and $\hat{\mathcal{S}}^\perp(\mathcal{D})$, respectively. Consider the singular value decompositions $\Pi_{\hat{\mathcal{S}}} M = U_{\hat{\mathcal{S}}} \Sigma_{\hat{\mathcal{S}}} V_{\hat{\mathcal{S}}}^\top$ and $\Pi_{\hat{\mathcal{S}}^\perp} M = U_{\hat{\mathcal{S}}^\perp} \Sigma_{\hat{\mathcal{S}}^\perp} V_{\hat{\mathcal{S}}^\perp}^\top$ Then, define

$$\hat{S} = U_{\hat{\mathcal{S}}}, \quad \hat{R} = U_{\hat{\mathcal{S}}^\perp}.$$

  The subspaces $\text{range}\,(\Pi_{\hat{\mathcal{S}}} M)$ and $\text{range}\,(\Pi_{\hat{\mathcal{S}}^\perp} M)$ are minimal subspaces contained in $\hat{\mathcal{S}}$ and $\hat{\mathcal{S}}^\perp$, respectively, such that $\text{range}\,(M) \subset \text{range}\,(\Pi_{\hat{\mathcal{S}}} M) \oplus \text{range}\,(\Pi_{\hat{\mathcal{S}}^\perp} M)$. The matrices $\hat{S}\hat{S}^\top$ and $\hat{R}\hat{R}^\top$ are their corresponding projection matrices.

- The **identified causal parameter** $\hat{\beta}^{\mathcal{S}}$ (approximately) equals the true causal parameter $\beta^\star$ on the space of training shift directions $\hat{\mathcal{S}}$. As conjectured in the anchor regression literature [41, 45, 23] (see, for example, the discussion right after Theorem 3.4 in [23] and Appendix H.3 therein) for $\gamma \to \infty$, the estimators $\beta_{\text{anchor}}^\gamma$ and $\beta_{\text{DRIG}}^\gamma$ converge to the causal parameter $\beta^\star$ on $\mathcal{S}$. Thus, the identified causal parameter can be estimated as

$$\hat{\beta}^{\mathcal{S}} := \Pi_{\hat{\mathcal{S}}} \beta_{\text{anchor}}^\infty \quad \text{or} \quad \hat{\beta}^{\mathcal{S}} := \Pi_{\hat{\mathcal{S}}} \beta_{\text{DRIG}}^\infty$$

  for the setting of mean or mean+variance shifts, respectively.

## D.2 Consistency of the worst-case robust predictor

For any estimator $\beta \in \mathbb{R}^d$ and given the estimated nuisance parameters $\hat{\varphi} := (\hat{S}\hat{S}^\top, \hat{R}\hat{R}^\top, \hat{\beta}^{\mathcal{S}})$, we define the sample worst-case robust risk as

$$\mathcal{L}_n(\beta, \hat{\varphi}) := \frac{1}{n_0} \sum_{i \in \mathcal{D}_0} \left( Y_{0,i} - \beta^\top X_{0,i} \right)^2 + \gamma \| \hat{S}\hat{S}^\top (\hat{\beta}^{\mathcal{S}} - \beta) \|_2^2 + \gamma \left( \sqrt{C - \|\hat{\beta}^{\mathcal{S}}\|_2^2} + \|\hat{R}\hat{R}^\top \beta\|_2 \right)^2. \tag{20}$$

Correspondingly, we define the estimator of the worst-case robust predictor by

$$\hat{\beta}_{\Theta_{\text{eq}}}^{\text{rob}} := \underset{\beta \in \mathcal{B}}{\arg\min}\, \mathcal{L}_n(\beta, \hat{\varphi}), \tag{21}$$

where $\mathcal{B} \subseteq \mathbb{R}^d$ is some compact set whose interior contains the true parameter $\beta_{\Theta_{\text{eq}}}^{\text{rob}}$.

To show the consistency of (21), we first require consistency of the nuisance parameter estimators, which we state as an assumption.

---

[12]In general, $S(\mathcal{D})$ is a proper subspace of $\mathbb{R}^d$ and the RHS of (19) corresponds to a sum of low-rank second moments. This can be consistently estimated if, for instance, the rank of each shift is known (e.g. in the mean shift setting), or the covariances have a spiked structure, allowing to cut off small eigenvalues.

**Assumption D.1.** *The estimated nuisance parameters $\hat{\varphi} := (\hat{S}\hat{S}^\top, \hat{R}\hat{R}^\top, \hat{\beta}^{\mathcal{S}})$ are consistent, that is, for $n \to \infty$,*

$$\|\hat{S}\hat{S}^\top - SS^\top\|_F \overset{\mathbb{P}}{\to} 0, \quad \|\hat{R}\hat{R}^\top - RR^\top\|_F \overset{\mathbb{P}}{\to} 0, \quad \hat{\beta}^{\mathcal{S}} \overset{\mathbb{P}}{\to} \beta^{\mathcal{S}} := \Pi_{\mathcal{S}}\beta^\star,$$

*where for any matrix $A \in \mathbb{R}^{m \times q}$, $\|A\|_F = \sqrt{\mathrm{tr}\,(A^\top A)}$ denotes the Frobenius norm, and $SS^\top$, $RR^\top$ are the corresponding population projection matrices onto $\Pi_{\mathcal{S}}\mathcal{M}$, $\Pi_{\mathcal{S}\perp}\mathcal{M}$ respectively.*

Depending on the assumptions of the data-generating process, Assumption D.1 can be shown to hold. For example, in the anchor regression setting [41], the consistency of the projection matrices $\hat{S}\hat{S}^\top$, $\hat{R}\hat{R}^\top$, and $\Pi_{\hat{\mathcal{S}}}$ holds if the dimension of $\mathcal{S}$ is known (due to the mean shift structure). The proof relies on the Davis–Kahan theorem (see, for example, [56]) and the consistency of the covariance matrix estimator. Moreover, in the anchor regression setting, it is conjectured that the estimator $\beta^\infty_{\text{anchor}}$ converges to its population counterpart (as discussed right after Theorem 3.4 in [23] and Appendix H.3 therein) which implies that $\hat{\beta}^{\mathcal{S}} := \Pi_{\hat{\mathcal{S}}}\beta^\infty_{\text{anchor}}$ consistently estimates $\beta^{\mathcal{S}} = \Pi_{\mathcal{S}}\beta^\star$.

Under the assumption of the consistency of the nuisance parameter estimators, we can now show that (21) is a consistent estimator of the worst-case robust predictor.

**Proposition 3.** *Consider the estimator $\hat{\beta}^{\mathrm{rob}}_{\Theta_{\mathrm{eq}}}$ of the worst-case robust predictor defined in (21). Suppose the optimization problem is over a compact set $\mathcal{B} \subseteq \mathbb{R}^d$ whose interior contains the true minimizer $\beta^{\mathrm{rob}}_{\Theta_{\mathrm{eq}}}$. Moreover, suppose Assumption D.1 holds. Finally, assume that the covariance matrix $\mathbb{E}\,[X_0 X_0^\top] \succ 0$ with bounded eigenvalues and $\mathbb{E}\,[Y_0^2] < \infty$. Then, $\hat{\beta}^{\mathrm{rob}}_{\Theta_{\mathrm{eq}}}$ is consistent, i.e., as $n, n_0 \to \infty$ it holds that*

$$\hat{\beta}^{\mathrm{rob}}_{\Theta_{\mathrm{eq}}} \overset{\mathbb{P}}{\to} \beta^{\mathrm{rob}}_{\Theta_{\mathrm{eq}}}.$$

### D.3  Proof of Proposition 3

For ease of notation define $\beta_0 := \beta^{\mathrm{rob}}_{\Theta_{\mathrm{eq}}}$ and $\hat{\beta} := \hat{\beta}^{\mathrm{rob}}_{\Theta_{\mathrm{eq}}}$. For any parameter of interest $\beta \in \mathcal{B}$ and nuisance parameters $\varphi = (P_S, P_R, b)$, define the function

$$(x, y) \mapsto g_{\beta,\varphi}(x, y) := (y - \beta^\top x)^2 + \gamma\|P_S(b - \beta)\|_2^2 + \gamma\left(\sqrt{C - \|b\|_2^2} + \|P_R\beta\|_2\right)^2. \quad (22)$$

Using (22), the robust identifiable risk and its sample version defined in (20) can be written, respectively as

$$\mathcal{L}(\beta, \varphi) = \mathbb{E}\,[g_{\beta,\varphi}(X_0, Y_0)], \quad \mathcal{L}_n(\beta, \varphi) = \frac{1}{n_0}\sum_{i \in \mathcal{D}_0} g_{\beta,\varphi}(X_{0,i}, Y_{0,i}).$$

Our goal is to show that $\hat{\beta} \overset{\mathbb{P}}{\to} \beta_0$. First, we show that the minimum of the loss is well-separated.

**Lemma D.1.** *Suppose that $\mathbb{E}\,[X_0 X_0^\top] \succ 0$. Then, for all $\delta > 0$, it holds that*

$$\inf\{\mathcal{L}(\beta, \varphi_0)\colon \|\beta - \beta_0\|_2 > \delta\} > \mathcal{L}(\beta_0, \varphi_0). \quad (23)$$

Fix $\delta > 0$. From the well-separation of the minimum from Lemma D.1, there exists $\varepsilon > 0$ such that

$$\left\{\|\hat{\beta} - \beta_0\|_2 > \delta\right\} \subseteq \left\{\mathcal{L}(\hat{\beta}, \varphi_0) - \mathcal{L}(\beta_0, \varphi_0) > \varepsilon\right\}.$$

Therefore,

$$\mathbb{P}\left(\|\hat{\beta} - \beta_0\|_2 > \delta\right) \leq \mathbb{P}\left(\mathcal{L}(\hat{\beta}, \varphi_0) - \mathcal{L}(\beta_0, \varphi_0) > \varepsilon\right)$$

$$= \mathbb{P}\Big(\mathcal{L}(\hat{\beta}, \varphi_0) - \mathcal{L}_n(\hat{\beta}, \varphi_0) + \mathcal{L}_n(\hat{\beta}, \varphi_0) - \mathcal{L}_n(\hat{\beta}, \hat{\varphi})$$

$$+ \mathcal{L}_n(\hat{\beta}, \hat{\varphi}) - \mathcal{L}_n(\beta_0, \hat{\varphi}) + \mathcal{L}_n(\beta_0, \hat{\varphi}) - \mathcal{L}(\beta_0, \varphi_0) > \varepsilon\Big)$$

$$\leq \mathbb{P}\left(\mathcal{L}(\hat{\beta}, \varphi_0) - \mathcal{L}_n(\hat{\beta}, \varphi_0) > \varepsilon/4\right) + \mathbb{P}\left(\mathcal{L}_n(\hat{\beta}, \varphi_0) - \mathcal{L}_n(\hat{\beta}, \hat{\varphi}) > \varepsilon/4\right) \quad (24)$$

$$+ \mathbb{P}\left(\mathcal{L}_n(\hat{\beta}, \hat{\varphi}) - \mathcal{L}_n(\beta_0, \hat{\varphi}) > \varepsilon/4\right) + \mathbb{P}\left(\mathcal{L}_n(\beta_0, \hat{\varphi}) - \mathcal{L}(\beta_0, \varphi_0) > \varepsilon/4\right). \quad (25)$$

We now want to prove convergence the four terms in (24) and (25). For this, we use the following statements proved in Appendix D.4.

**Lemma D.2.** *Suppose $\mathcal{B} \subseteq \mathbb{R}^d$ is a compact set. Moreover, assume that the covariance matrix $\mathbb{E}\left[X_0 X_0^\top\right] \succ 0$ with bounded eigenvalues and $\mathbb{E}\left[Y_0^2\right] < \infty$. Then, as $n, n_0 \to \infty$ it holds that*

$$\sup_{\beta \in \mathcal{B}} |\mathcal{L}_n(\beta, \varphi_0) - \mathcal{L}(\beta, \varphi_0)| \xrightarrow{\mathbb{P}} 0. \tag{26}$$

**Lemma D.3.** *As $n \to \infty$, it holds that*

$$\sup_{\beta \in \mathcal{B}} |\mathcal{L}_n(\beta, \hat{\varphi}) - \mathcal{L}_n(\beta, \varphi_0)| \xrightarrow{\mathbb{P}} 0. \tag{27}$$

The two terms in (24) converge to 0 by Lemma D.2 and Lemma D.3, respectively. The first term in (25) equals 0 since $\hat{\beta}$ minimizes $\beta \mapsto \mathcal{L}_n(\beta, \hat{\varphi})$. Finally, we observe that

$$\sup_{\beta \in \mathcal{B}} |\mathcal{L}_n(\beta, \hat{\varphi}) - \mathcal{L}(\beta, \varphi_0)| \xrightarrow{\mathbb{P}} 0, \tag{28}$$

since we have that

$$\sup_{\beta \in \mathcal{B}} |\mathcal{L}_n(\beta, \hat{\varphi}) - \mathcal{L}(\beta, \varphi_0)| \leq \sup_{\beta \in \mathcal{B}} |\mathcal{L}_n(\beta, \hat{\varphi}) - \mathcal{L}_n(\beta, \varphi_0)| + \sup_{\beta \in \mathcal{B}} |\mathcal{L}_n(\beta, \varphi_0) - \mathcal{L}(\beta, \varphi_0)|,$$

where the first term converges in probability by Lemma D.3, and the second term converges in probability by Lemma D.2. This implies that the second term in (25) converges to zero. Since $\delta > 0$ was arbitrary, it follows that $\hat{\beta} \xrightarrow{\mathbb{P}} \beta_0$.

### D.4 Proof of auxiliary lemmas

#### D.4.1 Proof of Lemma D.1

By definition,

$$\mathcal{L}(\beta, \varphi_0) = \mathbb{E}\left[(Y_0 - \beta^\top X_0)^2\right] + \gamma \|SS^\top(\beta^\mathcal{S} - \beta)\|_2^2 + \gamma \left(\sqrt{C - \|\beta^\mathcal{S}\|_2^2} + \|RR^\top \beta\|_2\right)^2.$$

Since $\mathbb{E}\left[X_0 X_0^\top\right] \succ 0$, the first term is strongly convex in $\beta$. Moreover, the second and third terms are convex in $\beta$. Therefore, $\mathcal{L}(\beta, \varphi_0)$ is strongly convex in $\beta$. Since $\mathcal{L}(\beta, \varphi_0)$ is also continuous in $\beta$, it follows that there exists a unique global minimum. Let $\beta_0$ denote the global minimizer of $\mathcal{L}(\beta, \varphi_0)$. By the fact that $\mathcal{L}(\beta_0, \varphi_0)$ is a global minimum, and by definition of strong convexity, there exists a positive constant $m > 0$ such that, for all $\beta \in \mathcal{B}$,

$$\mathcal{L}(\beta, \varphi_0) \geq \mathcal{L}(\beta_0, \varphi_0) + \frac{m}{2} \|\beta - \beta_0\|_2^2. \tag{29}$$

Fix $\delta > 0$. Then, by (29), for all $\beta \in \mathcal{B}$ such that $\|\beta - \beta_0\|_2 > \delta$ it holds that

$$\mathcal{L}(\beta, \varphi_0) \geq \mathcal{L}(\beta_0, \varphi_0) + \frac{m\delta^2}{2} > \mathcal{L}(\beta_0, \varphi_0).$$

Since the inequality holds for all $\beta \in \mathcal{B}$ such that $\|\beta - \beta_0\|_2 > \delta$, we conclude that

$$\inf\{\mathcal{L}(\beta, \varphi_0) \colon \|\beta - \beta_0\|_2 > \delta\} > \mathcal{L}(\beta_0, \varphi_0).$$

Since $\delta > 0$ was arbitrary, the claim follows.

#### D.4.2 Proof of Lemma D.2

Recall that for any $\beta \in \mathcal{B}$

$$\mathcal{L}(\beta, \varphi_0) = \mathbb{E}\left[g_{\beta, \varphi_0}(X_0, Y_0)\right], \quad \mathcal{L}_n(\beta, \varphi_0) = \frac{1}{n_0} \sum_{i \in \mathcal{D}_0} g_{\beta, \varphi_0}(X_{0,i}, Y_{0,i}).$$

To show the result, we must establish that the class of functions $\{g_{\beta, \varphi_0} \colon \beta \in \mathcal{B}\}$ is Glivenko–Cantelli. From [53], a set of sufficient conditions for being a Glivenko–Cantelli class is that (i) $\mathcal{B}$ is compact, (ii) $\beta \mapsto g_{\beta, \varphi_0}(x, y)$ is continuous for every $(x, y)$, and (iii) $\beta \mapsto g_{\beta, \varphi_0}$ is dominated by an integrable function. By assumption, (i) holds. Moreover, by (22), it follows that $\beta \mapsto g_{\beta, \varphi_0}$ is continuous

for all $(x, y)$ and thus (ii) holds. We now show that (iii) holds. Since $\mathcal{B}$ is compact we have that $\sup_{\beta \in \mathcal{B}} \|\beta\|_2 = C_1 < \infty$. For fixed $\gamma > 0$, and all $(x, y)$, we have that

$$
\begin{aligned}
g_{\beta, \varphi_0}(x, y) &\leq \sup_{\beta \in \mathcal{B}} |g_{\beta, \varphi_0}(x, y)| \\
&\leq \sup_{\beta \in \mathcal{B}} (y - \beta^\top x)^2 + 2\gamma \|SS^\top\|_F^2 \left( \|\beta^{\mathcal{S}}\|_2^2 + \sup_{\beta \in \mathcal{B}} \|\beta\|_2^2 \right) \\
&\quad + \gamma \left( \sqrt{C - \|\beta^{\mathcal{S}}\|_2^2} + \|RR^\top\|_F \sup_{\beta \in \mathcal{B}} \|\beta\|_2 \right)^2 \\
&\leq 2y^2 + 2C_1^2 \|x\|_2^2 + K =: G(x, y),
\end{aligned}
\tag{30}
$$

where $K < \infty$ is a finite constant not depending on $(x, y)$. Furthermore, we have that

$$
\mathbb{E}\left[ G(X_0, Y_0) \right] = 2\mathbb{E}\left[ Y_0^2 \right] + 2C_1^2 \operatorname{tr}\left( \mathbb{E}\left[ X_0 X_0^\top \right] \right) + K < \infty,
\tag{31}
$$

since $\mathbb{E}\left[ Y^2 \right] < \infty$ and $\mathbb{E}\left[ X_0 X_0^\top \right]$ has bounded eigenvalues by assumption. From (30) and (31), it follows that (iii) holds.

### D.4.3  Proof of Lemma D.3

For fixed $\gamma > 0$, we have that

$$
\frac{1}{\gamma} \sup_{\beta \in \mathcal{B}} |\mathcal{L}_n(\beta, \hat{\varphi}) - \mathcal{L}_n(\beta, \varphi_0)| \leq \sup_{\beta \in \mathcal{B}} \left| \|\hat{S}\hat{S}^\top(\hat{\beta}^{\mathcal{S}} - \beta)\|_2^2 - \|SS^\top(\beta^{\mathcal{S}} - \beta)\|_2^2 \right|
\tag{32}
$$

$$
+ \sup_{\beta \in \mathcal{B}} \left| \left( \sqrt{C - \|\hat{\beta}^{\mathcal{S}}\|_2^2} + \|\hat{R}\hat{R}^\top \beta\|_2 \right)^2 - \left( \sqrt{C - \|\beta^{\mathcal{S}}\|_2^2} + \|RR^\top \beta\|_2 \right)^2 \right|
\tag{33}
$$

We can upper bound (32) as follows,

$$
\begin{aligned}
&\sup_{\beta \in \mathcal{B}} \left| \|\hat{S}\hat{S}^\top(\hat{\beta}^{\mathcal{S}} - \beta)\|_2^2 - \|SS^\top(\beta^{\mathcal{S}} - \beta)\|_2^2 \right| \\
&= \sup_{\beta \in \mathcal{B}} \left| (\hat{\beta}^{\mathcal{S}} - \beta)^\top \hat{S}\hat{S}^\top (\hat{\beta}^{\mathcal{S}} - \beta) - (\beta^{\mathcal{S}} - \beta)^\top SS^\top (\beta^{\mathcal{S}} - \beta) \right| \\
&= \sup_{\beta \in \mathcal{B}} \left| (\hat{\beta}^{\mathcal{S}} - \beta)^\top \hat{S}\hat{S}^\top (\hat{\beta}^{\mathcal{S}} - \beta^{\mathcal{S}}) + (\hat{\beta}^{\mathcal{S}} - \beta^{\mathcal{S}})^\top \hat{S}\hat{S}^\top (\beta^{\mathcal{S}} - \beta) \right. \\
&\quad \left. + (\beta^{\mathcal{S}} - \beta)^\top (\hat{S}\hat{S}^\top - SS^\top)(\beta^{\mathcal{S}} - \beta) \right| \\
&\leq 2 \sup_{\beta \in \mathcal{B}} \|\hat{\beta}^{\mathcal{S}} - \beta\|_2 \, \|\hat{S}\hat{S}^\top\|_F \, \|\hat{\beta}^{\mathcal{S}} - \beta^{\mathcal{S}}\|_2 + \sup_{\beta \in \mathcal{B}} \|\beta^{\mathcal{S}} - \beta\|_2^2 \, \|\hat{S}\hat{S}^\top - SS^\top\|_F
\tag{34}
\\
&\leq C_1 \|\hat{\beta}^{\mathcal{S}} - \beta^{\mathcal{S}}\|_2 + C_2 \|\hat{S}\hat{S}^\top - SS^\top\|_F \xrightarrow{\mathbb{P}} 0,
\tag{35}
\end{aligned}
$$

where (34) follows from the Cauchy–Schwarz inequality and that $\|A\|_2 \leq \|A\|_F$, the constants $C_1, C_2 < \infty$ in (35) follow from compactness of $\mathcal{B}$, and the convergence in probability follows from Assumption D.1. Furthermore, we can upper bound (33) as follows,

$$
\begin{aligned}
&\sup_{\beta \in \mathcal{B}} \left| \left( \sqrt{C - \|\hat{\beta}^{\mathcal{S}}\|_2^2} + \|\hat{R}\hat{R}^\top \beta\|_2 \right)^2 - \left( \sqrt{C - \|\beta^{\mathcal{S}}\|_2^2} + \|RR^\top \beta\|_2 \right)^2 \right| \\
&= \sup_{\beta \in \mathcal{B}} \left| C - \|\hat{\beta}^{\mathcal{S}}\|_2^2 + \|\hat{R}\hat{R}^\top \beta\|_2^2 + 2\sqrt{C - \|\hat{\beta}^{\mathcal{S}}\|_2^2} \, \|\hat{R}\hat{R}^\top \beta\|_2 \right. \\
&\quad \left. - C + \|\beta^{\mathcal{S}}\|_2^2 - \|RR^\top \beta\|_2^2 - 2\sqrt{C - \|\beta^{\mathcal{S}}\|_2^2} \, \|RR^\top \beta\|_2 \right| \\
&\leq \sup_{\beta \in \mathcal{B}} \left| \|\hat{\beta}^{\mathcal{S}}\|_2^2 - \|\beta^{\mathcal{S}}\|_2^2 \right| + \sup_{\beta \in \mathcal{B}} \left| \beta^\top (\hat{R}\hat{R}^\top - RR^\top)\beta \right| \\
&\quad + 2 \sup_{\beta \in \mathcal{B}} \left| \sqrt{C - \|\hat{\beta}^{\mathcal{S}}\|_2^2} \, \|\hat{R}\hat{R}^\top \beta\|_2 - \sqrt{C - \|\beta^{\mathcal{S}}\|_2^2} \, \|RR^\top \beta\|_2 \right| \\
&= (I) + (II) + (III).
\end{aligned}
$$

By Assumption D.1, $(I)$ converges in probability to zero. Regarding $(II)$, we have

$$\sup_{\beta \in \mathcal{B}} \left| \beta^\top (\hat{R}\hat{R}^\top - RR^\top)\beta \right| \leq \sup_{\beta \in \mathcal{B}} \|\beta\|_2^2 \, \|\hat{R}\hat{R}^\top - RR^\top\|_F \xrightarrow{\mathbb{P}} 0,$$

where the inequality follows from Cauchy–Schwarz and that $\|A\|_2 \leq \|A\|_F$, and the convergence in probability follows from Assumption D.1 along with the compactness of $\mathcal{B}$. It remains to upper bound $(III)$. We have that

$$\frac{(III)}{2} \leq \sup_{\beta \in \mathcal{B}} \left| \sqrt{C - \|\hat{\beta}^{\mathcal{S}}\|_2^2} \, \|\hat{R}\hat{R}^\top \beta\|_2 - \sqrt{C - \|\beta^{\mathcal{S}}\|_2^2} \, \|\hat{R}\hat{R}^\top \beta\|_2 \right|$$

$$+ \sup_{\beta \in \mathcal{B}} \left| \sqrt{C - \|\beta^{\mathcal{S}}\|_2^2} \, \|\hat{R}\hat{R}^\top \beta\|_2 - \sqrt{C - \|\beta^{\mathcal{S}}\|_2^2} \, \|RR^\top \beta\|_2 \right|$$

$$\leq \left( \sup_{\beta \in \mathcal{B}} \|\beta\|_2 \, \|\hat{R}\hat{R}^\top\|_F \right) \left| \sqrt{C - \|\hat{\beta}^{\mathcal{S}}\|_2^2} - \sqrt{C - \|\beta^{\mathcal{S}}\|_2^2} \right|$$

$$+ \sup_{\beta \in \mathcal{B}} \left| \sqrt{\beta^\top \hat{R}\hat{R}^\top \beta} - \sqrt{\beta^\top RR^\top \beta} \right| \left( \sqrt{C - \|\beta^{\mathcal{S}}\|_2^2} \right)$$

$$\leq C_3 \left| \|\beta^{\mathcal{S}}\|_2^2 + \|\hat{\beta}^{\mathcal{S}}\|_2^2 \right|^{1/2} + \sqrt{C} \sup_{\beta \in \mathcal{B}} \left| \beta^\top (\hat{R}\hat{R}^\top - RR^\top)\beta \right|^{1/2} \qquad (36)$$

$$\leq C_3 \left| \|\beta^{\mathcal{S}}\|_2^2 + \|\hat{\beta}^{\mathcal{S}}\|_2^2 \right|^{1/2} + \sqrt{C} \left( \sup_{\beta \in \mathcal{B}} \|\beta\|_2^2 \, \|\hat{R}\hat{R}^\top - RR^\top\|_F \right)^{1/2} \xrightarrow{\mathbb{P}} 0. \qquad (37)$$

The inequality in (36) follows from the compactness of $\mathcal{B}$, the fact that $\hat{R}\hat{R}^\top$ has bounded eigenvalues, and that $|\sqrt{x} - \sqrt{y}| \leq |x - y|^{1/2}$ for all $x, y \geq 0$. The inequality in (37) follows from Cauchy–Schwarz and that $\|A\|_2 \leq \|A\|_F$. The convergence in probability follows form Assumption D.1 and the compactness of $\mathcal{B}$.

# E  Details on finite-sample experiments

In this section, we provide more details of the data generation for our synthetic finite-sample experiments as well as data processing for the real-world data experiments.

## E.1  Synthetic experiments

For the synthetic experiments, we generate a random SCM which satisfies our assumptions. For $d = 15$, we randomly sample the joint covariance $\Sigma^\star$ of $(\eta, \xi)$, fixing its total variance and the eigenvalues. We consider 7 environments including the reference environment, and for each environment except the reference, we randomly generate mean shifts $\mu_e$ of fixed norm 1. Since we have 6 non-zero random Gaussian mean shifts, it holds a.s. that $\dim \mathcal{S} = 6$. We then randomly generate an "initial guess" for $\beta^\star \in \mathbb{R}^d$ of fixed norm $C = 10$. Now, with respect to the space $\mathcal{S}$ of the identifiable directions induced by the mean shifts, we choose the most "adversarial" causal parameter $\beta_{\mathrm{adv}}^\star$ which is equal to $\beta^\star$ on $\mathcal{S}$, but on $\mathcal{S}^\perp$ has the opposite direction of the noise OLS estimator $\Sigma_\eta^{\star-1}\Sigma_{\eta,\xi}^\star$. We ensure that $\|\beta_{\mathrm{adv}}^\star\|_2 = C$. Note that under the observed shifts, $\beta^\star$ and $\beta_{\mathrm{adv}}^\star$ are observationally equivalent. We complete $\beta_{\mathrm{adv}}^\star$ to the set $\theta_{\mathrm{adv}}$ of observationally equivalent model parameters and generate the multi-environment training data according to $\theta_{\mathrm{adv}}$ and the collection of mean shifts.

For Figure 3 (left), we define the test shift upper bound as $M_{\mathrm{anchor}} = \gamma \frac{1}{7} \sum_e \mu_e \mu_e^\top$. We vary $\gamma$ from 0 to 10, and for each $\gamma$, we compute the oracle anchor regression estimator by minimizing the discrete anchor regression loss with the correct $\gamma$. Additionally, we compute the pooled OLS estimator and the worst-case robust predictor $\beta_{\Theta_{\mathrm{eq}}}^{\mathrm{rob}}$ as described in Appendix D. Finally, we generate test data with a Gaussian additive shift $A_{\mathrm{test}} \sim \mathcal{N}(0, M_{\mathrm{anchor}})$. We evaluate the loss of $\beta_{\mathrm{OLS}}$, $\beta_{\mathrm{anchor}}$ and $\beta_{\Theta_{\mathrm{eq}}}^{\mathrm{rob}}$ on this test environment and include the population lower bound.

For Figure 3 (right), we define the test shift upper bound as $M_{\mathrm{new}} = \gamma \frac{1}{7} \sum_e \mu_e \mu_e^\top + \gamma' RR^\top$, where $R$ is a 2-dimensional subspace of the space $\mathcal{S}^\perp$. We fix the magnitude $\gamma$ of the "seen" test shift

directions at $\gamma = 40$ and set vary $\gamma'$ from 0 to 2 to showcase the effect of small unseen shifts compared to large identified shifts. We compute the oracle anchor regression estimator by minimizing the discrete anchor regression loss. Additionally, we compute the pooled OLS estimator and the worst-case robust predictor $\beta_{\Theta_{\mathrm{eq}}}^{\mathrm{rob}}$ as described in Appendix D, for which we use the oracle $\gamma'$, given $M_{\mathrm{anchor}}$ and empirical estimates of the spaces $\mathcal{S}, \mathcal{S}^{\perp}, R$.

Finally, we generate test data with a Gaussian additive shift $A_{\mathrm{test}} \sim \mathcal{N}(0, M_{\mathrm{new}})$. We evaluate the loss of $\beta_{\mathrm{OLS}}, \beta_{\mathrm{anchor}}$ and $\beta_{\Theta_{\mathrm{eq}}}^{\mathrm{rob}}$ on this test environment, plot the resulting test losses for different estimators and include the population lower bound.

## E.2 Real-world data experiments

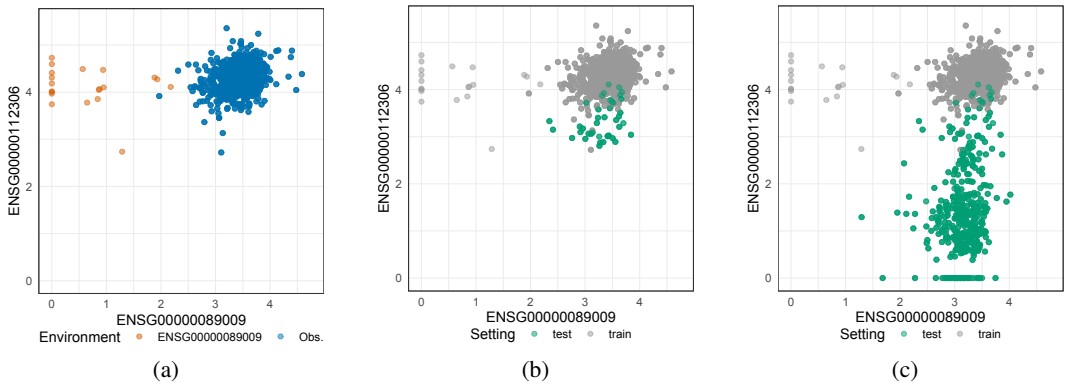

(a)  (b)  (c)

Figure 5: The figures illustrate the structure of the (a) training-time shifts and (b-c) test-time shifts for different perturbation strengths on the example of two covariates. Panel (a) shows the training data containing two environments–observational (blue) and shifted (orange) corresponding to the knockout of the gene ENSG00000089009. Panels (b) and (c) show the training data in grey and test data from a previously unseen environment (green). Panel (b) depicts the top $10\%$ test data points closest to the training support (perturbation strength = 0.1). Panel (c) illustrates the full test data (perturbation strength = 1.0).

We consider the K562 dataset from [38] and perform the preprocessing as done in [13]. The resulting dataset consists of $n = 162,751$ single-cell observations over $d = 622$ genes collected from observational and several interventional environments. The interventional environments arise by knocking down a single gene at a time using the CRISPR interference method [37]. Following [44], we select only always-active genes in the observational setting, resulting in a smaller dataset of 28 genes. For each gene $j = 1, \ldots, 28$, we set $Y := X_j$ as the target variable and select the three genes $X_{k_1}, \ldots, X_{k_3}$ most strongly correlated with $Y$ (using Lasso), resulting in a prediction problem over $Y, X_{k_1}, \ldots, X_{k_3}$. Given this prediction problem, we construct the training and test datasets as follows. Let $\mathcal{O}$ denote the 10,691 observations collected from the observational environment, and let $\mathcal{I}_i$ denote the observations collected from the interventional environment where the gene $k_i$ was knocked down. We will denote by $\mathcal{I}_{i,s}$ the $s \times 100$ percent of datapoints in $\mathcal{I}_i$ that are closest to the mean of gene $k_i$ in the observational environment $\mathcal{O}$. For example, $\mathcal{I}_{i,0.1}$ consists of the 10% of datapoints in $\mathcal{I}_i$ closest to the observational mean of gene $k_i$. Thus, the parameter $s \in [0, 1]$ acts as a proxy for the *strength* of the shift. Denote by $\mathcal{I}_{i,s}^*$ a random sample of $\mathcal{I}_{i,s}$ of a certain size. For each $i \in \{1, 2, 3\}$, we fit the methods on the training data $\mathcal{D}_i^{\mathrm{train}} := \mathcal{O} \cup \mathcal{I}_{i,1}^*$, with $|\mathcal{I}_{i,1}^*| = 20$. Figure 5(a) illustrates an example of training data $\mathcal{D}_i^{\mathrm{train}}$. Having fitted the methods on $\mathcal{D}_i^{\mathrm{train}}$, we evaluate them on test datasets constructed as follows. For each shift strength $s \in \{0.1, \ldots, 0.9\}$ and proportion $\pi \in \{0, .33, .67, 1\}$, define the test dataset $\mathcal{D}_{\pi,s}$ consisting of $\pi$ observations from $\cup_{\ell \neq i} \mathcal{I}_{\ell,s}$ and $1 - \pi$ (out-of-training) observations from $\mathcal{I}_{i,s}$. An example of a test dataset for different shift strengths $s$ and previously unseen directions (i.e., $\pi = 1$) is shown in Figure 5(b-c). We compare our method Worst-case Rob., defined as the minimizer of the empirical worst-case robust risk (20), with anchor regression [41], invariant causal prediction (ICP) [35], Distributional Robustness via Invariant Gradients (DRIG) [45], and OLS (corresponding to vanilla ERM). We use the following parameters for Worst-case Rob.: $\gamma = 50$, $C_{\mathrm{ker}} = 1.0$, and $M = \mathrm{Id}$. For anchor regression and DRIG, we select $\gamma = 50$. For ICP, we set the significance level for the invariance tests to $\alpha = 0.05$.

These numerical experiments are computationally light and can be run in $\approx 5$ minutes on a personal laptop.[13]

# F  Proofs

## F.1  Proof of Proposition 1

For every environment $e \in \mathcal{E}_{\text{train}}$, we observe the first moments $\mathbb{E}(X_e)$ and $\mathbb{E}(Y_e)$, and second moments $\mathbb{E}(X_e X_e^\top)$, $\mathbb{E}(Y_e^2)$ and $\mathbb{E}(X_e Y_e)$. Since it holds by assumption that $\mu_0 = 0$ and $\Sigma_0 = 0$, we have that $\mathbb{E}(X_0 X_0^\top) = \Sigma_\eta^\star$, and so we can identify $\Sigma_\eta^\star$ uniquely. Furthermore, it holds that

$$\mathbb{E}(X_0 Y_0) = \Sigma_\eta^\star \beta^\star + \Sigma_{\eta,\xi}^\star, \tag{38}$$

$$\mathbb{E}(X_e Y_e) = (\Sigma_e + \mu_e \mu_e^\top + \Sigma_\eta^\star)\beta^\star + \Sigma_{\eta,\xi}^\star. \tag{39}$$

By taking the difference between Equation (39) and Equation (38), we can identify $(\Sigma_e + \mu_e \mu_e^\top)\beta^\star$. Thus, the parameter $\beta^\star$ is identifiable on the subspace $\mathcal{S}$ defined in Equation (12) and is not identifiable on its orthogonal complement $\mathcal{S}^\perp$. Thus, for any vector $\alpha \in \mathcal{S}^\perp$, the vector $\beta = \beta^\star + \alpha$ is consistent with the data-generating process. It remains to compute the covariance parameters induced by an arbitrary $\tilde\beta := \beta^\star + \alpha$, for $\alpha \in \mathcal{S}^\perp$. For every environment $e \in \mathcal{E}_{\text{train}}$, the second mixed moment between $X_e$ and $Y_e$ has to satisfy the following equality

$$\mathbb{E}(X_e Y_e) = (\Sigma_e + \mu_e \mu_e^\top + \Sigma_\eta^\star)\beta^\star + \Sigma_{\eta,\xi}^\star = (\Sigma_e + \mu_e \mu_e^\top + \Sigma_\eta^\star)\tilde\beta + \tilde\Sigma_{\eta,\xi},$$

from which it follows that $\tilde\Sigma_{\eta,\xi} := \Sigma_{\eta,\xi}^\star - \Sigma_\eta^\star \alpha$. By computing $\mathbb{E}(Y_e^2)$ and inserting $\tilde\beta = \beta^\star + \alpha$ and $\tilde\Sigma_{\eta,\xi}$, we similarly obtain

$$\tilde\sigma_\xi^2 := (\sigma_\xi^\star)^2 - 2\alpha^\top \Sigma_{\eta,\xi}^\star + \alpha^\top \Sigma_\eta^\star \alpha.$$

Thus, we obtain the following set of observationally equivalent model parameters consistent with $\mathcal{P}_{\theta^\star, \mathcal{E}_{\text{train}}}$:

$$\Theta_{\text{eq}} = \{\beta^\star + \alpha, \Sigma_\eta^\star, \Sigma_{\eta,\xi}^\star - \Sigma_\eta^\star \alpha, (\sigma_\xi^\star)^2 - 2\alpha^\top \Sigma_{\eta,\xi}^\star + \alpha^\top \Sigma_\eta^\star \alpha \colon \alpha \in \mathcal{S}^\perp\}.$$

Since the observationally equivalent set is identifiable from the training distribution, but model parameters $\beta^\star, \Sigma_{\eta,\xi}^\star, (\sigma_\xi^\star)^2$ are not, it is helpful to re-express the observationally equivalent set through identifiable quantities. For this, we note that the "identifiable linear predictor" $\beta^\mathcal{S} = \beta^\star - \beta^{\mathcal{S}^\perp}$ induces an observationally equivalent model given by

$$\theta^\mathcal{S} := (\beta^\mathcal{S}, \Sigma_\eta^\mathcal{S}, \Sigma_{\eta,\xi}^\mathcal{S}, (\sigma_\xi^\mathcal{S})^2) = (\beta^\mathcal{S}, \Sigma_\eta^\star, \Sigma_{\eta,\xi}^\star + \Sigma_\eta^\star \beta^{\mathcal{S}^\perp}, (\sigma_\xi^\star)^2 + 2\langle \Sigma_{\eta,\xi}^\star, \beta^{\mathcal{S}^\perp}\rangle + \langle \beta^{\mathcal{S}^\perp}, \Sigma_\eta^\star \beta^{\mathcal{S}^\perp}\rangle).$$

From this reparameterization, we infer the final form of the observationally equivalent set:

$$\Theta_{\text{eq}} = \{\beta^\mathcal{S} + \alpha, \Sigma_\eta', \Sigma_{\eta,\xi}^\mathcal{S} - \Sigma_\eta' \alpha, (\sigma_\xi^\mathcal{S})^2 - 2\alpha^\top \Sigma_{\eta,\xi}^\mathcal{S} + \alpha^\top \Sigma_\eta' \alpha \colon \alpha \in \mathcal{S}^\perp\} \ni \theta^\star$$

Therefore, Equation (13) follows. To find the robust predictor $\beta^{rob}$, we write down the robust loss with respect to $M_{\text{test}}$ and any $\theta_\alpha$ from the observationally equivalent set:

$$\mathcal{R}_{\text{rob}}(\beta; \theta_\alpha, M_{\text{test}}) = (\beta^\mathcal{S} + \alpha - \beta)^\top (M_{\text{test}} + \Sigma_\eta^\star)(\beta^\mathcal{S} + \alpha - \beta)$$
$$+ 2(\beta^\mathcal{S} + \alpha - \beta)^\top (\Sigma_{\eta,\xi}^\star - \Sigma_\eta^\star \alpha) + (\sigma_\xi^\mathcal{S})^2 - 2\alpha^\top \Sigma_{\eta,\xi}^\mathcal{S} + \alpha^\top \Sigma_\eta^\star \alpha.$$

inserting $\alpha \in \mathcal{S}^\perp$ and rearranging, Equation (14) follows.

## F.2  Proof of Theorem 3.1

We structure the proof as follows: first, we quantify the non-identifiability of the robust risk by explicitly computing its supremum over the observationally equivalent set of the model parameters (referred to as the worst-case robust risk). Second, we derive a lower bound for the worst-case robust risk by considering two cases depending on how a predictor $\bar\beta$ interacts with the possible test shifts $M_{\text{test}}$.

---

[13]We use a 2020 13-inch MacBook Pro with a 1.4 GHz Quad-Core Intel Core i5 processor, 8 GB of RAM, and Intel Iris Plus Graphics 645 with 1536 MB of graphics memory.

**Computation of the worst-case robust risk.** For any model-generating parameter $\theta = (\beta, \Sigma)$ it holds that the robust risk of the model Equation (3) under test shifts $M_{\text{test}} \succeq 0$ is given by

$$\mathcal{R}_{\text{rob}}(\overline{\beta}; \theta, M_{\text{test}}) = (\beta - \overline{\beta})^\top (M_{\text{test}} + \Sigma_\eta^\star)(\beta - \overline{\beta}) + 2(\beta - \overline{\beta})^\top \Sigma_{\eta,\xi} + (\sigma_\xi)^2.$$

We recall that the observationally equivalent set of model parameters after observing the multi-environment training data Equation (3) is given by

$$\Theta_{\text{eq}} = \{\beta^{\mathcal{S}} + \alpha, \Sigma_\eta^\star, \Sigma_{\eta,\xi}^{\mathcal{S}} - \Sigma_\eta^\star \alpha, (\sigma_\xi^{\mathcal{S}})^2 - 2\alpha^\top \Sigma_{\eta,\xi}^{\mathcal{S}} + \alpha^\top \Sigma_\eta \alpha : \alpha \in \mathcal{S}^\perp\}, \tag{40}$$

where $\mathcal{S}$ is the span of identified directions defined in Equation (12). Moreover, we recall that by Assumption 3.2, for any causal parameter $\beta$ it should hold that $\|\beta\|_2 = \|\beta^{\mathcal{S}} + \alpha\|_2 \leq C$, which translates into the following constraint for the parameter $\alpha$:

$$\|\alpha\|_2 \leq \sqrt{C^2 - \|\beta^{\mathcal{S}}\|_2^2} =: C_{\text{ker}}.$$

Inserting Equation (40) in Equation (9), we obtain

$$\mathfrak{R}_{\text{rob}}(\overline{\beta}; \Theta_{\text{eq}}, M_{\text{test}}) = \sup_{\substack{\alpha \in \mathcal{S}^\perp, \\ \|\alpha\|_2 \leq C_{\text{ker}}}} \mathcal{R}_{\text{rob}}(\overline{\beta}; \theta_\alpha, M_{\text{test}}),$$

where $\theta_\alpha$ is a short notation for $(\beta^{\mathcal{S}} + \alpha, \Sigma_\eta^\star, \Sigma_{\eta,\xi}^{\mathcal{S}} - \Sigma_\eta^\star \alpha, (\sigma_\xi^{\mathcal{S}})^2 - 2\alpha^\top \Sigma_{\eta,\xi}^{\mathcal{S}} + \alpha^\top \Sigma_\eta^\star \alpha)$. We now compute the supremum explicitly in case $M_{\text{test}}$ has the form $M_{\text{test}} = \gamma M_{\text{seen}} + \gamma' RR^\top$, where $M_{\text{seen}}$ is a PSD matrix with range $M \subseteq \mathcal{S}$ and $R$ is a semi-orthogonal matrix with range $R \subseteq \mathcal{S}^\perp$. For any $\alpha \in \mathcal{S}^\perp$, we write down the robust loss as

$$\mathcal{R}_{\text{rob}}(\overline{\beta}; \theta_\alpha, M_{\text{test}}) = (\beta^{\mathcal{S}} - \overline{\beta})^\top (M_{\text{test}} + \Sigma_\eta^\star)(\beta^{\mathcal{S}} - \overline{\beta}) + 2(\beta^{\mathcal{S}} - \overline{\beta})^\top \Sigma_{\eta,\xi}^{\mathcal{S}} + (\sigma_\xi^{\mathcal{S}})^2$$
$$+ \alpha^\top M_{\text{test}} \alpha + 2\alpha^\top M_{\text{test}}(\beta^{\mathcal{S}} - \overline{\beta})$$
$$= \mathcal{R}_{\text{rob}}(\overline{\beta}; \theta^{\mathcal{S}}, M_{\text{test}}) + \alpha^\top M_{\text{test}} \alpha + 2\alpha^\top M_{\text{test}}(\beta^{\mathcal{S}} - \overline{\beta}).$$

The first term is the robust risk of $\overline{\beta}$ under test shift $M_{\text{test}}$ and the identified model-generating parameter $\theta^{\mathcal{S}}$, thus it does not depend on $\alpha$. By the structure of $M_{\text{test}}$, we obtain that

$$f(\alpha) := \alpha^\top M_{\text{test}} \alpha + 2\alpha^\top M_{\text{test}}(\beta^{\mathcal{S}} - \overline{\beta}) = \gamma' \alpha^\top RR^\top \alpha - \gamma' \alpha^\top RR^\top \overline{\beta}.$$

If $\gamma' = 0$, i.e., the test shifts consist only of the identified directions, we have $f(\alpha) = 0$, independently of $\alpha$, and thus

$$\mathfrak{R}_{\text{rob}}(\overline{\beta}; \Theta_{\text{eq}}, M_{\text{test}}) = \mathcal{R}_{\text{rob}}(\overline{\beta}; \theta^{\mathcal{S}}, M_{\text{test}}).$$

This implies the first statement of the theorem.

We now consider the case where $R \neq 0$, i.e., $RR^\top$ is a non-degenerate projection. Our goal is to maximize $f(\alpha)$ subject to constraints $\alpha \in \mathcal{S}^\perp$, $\|\alpha\|_2 \leq C_{\text{ker}}$. Let $\tilde{R}$ be an orthonormal extension of $R$ such that range $(R|\tilde{R}) = \mathcal{S}^\perp$. Then, we can parameterize $\alpha \in \mathcal{S}^\perp$ as $\alpha = (R|\tilde{R})(\frac{w}{\tilde{w}})$ and the corresponding Lagrangian reads

$$\mathcal{L}(\alpha, \lambda) = \gamma' \alpha^\top RR^\top \alpha - \gamma' \alpha^\top RR^\top \overline{\beta} + \lambda(C_{\text{ker}}^2 - \|\alpha\|_2^2)$$
$$= \gamma' \|w\|_2^2 - \gamma' w^\top R^\top \overline{\beta} + \lambda(C_{\text{ker}}^2 - \|(w, \tilde{w})\|_2^2).$$

Differentiating with respect to $w, \tilde{w}$ yields

$$w = \frac{\gamma'}{\gamma' - \lambda} R^\top \overline{\beta};$$
$$\tilde{w} = 0.$$

After differentiating w.r.t. $\lambda$, we obtain $\frac{\gamma'}{\gamma' - \lambda} = \pm \frac{C_{\text{ker}}}{\|R^\top \overline{\beta}\|_2}$. By inserting in the objective function and comparing, we obtain the **value of the worst-case robust risk**:

$$\mathfrak{R}_{\text{rob}}(\overline{\beta}; \Theta_{\text{eq}}, M_{\text{test}}) = \gamma' C_{\text{ker}}^2 + 2\gamma' \|R^\top \overline{\beta}\|_2 + \mathcal{R}_{\text{rob}}(\overline{\beta}; \theta^{\mathcal{S}}, M_{\text{test}}) \tag{41}$$
$$= \gamma' C_{\text{ker}}^2 + 2\gamma' \|R^\top \overline{\beta}\|_2 + \overline{\beta}^\top RR^\top \overline{\beta} + \gamma(\beta^{\mathcal{S}} - \beta)^\top M_{\text{seen}}(\beta^{\mathcal{S}} - \beta) + \mathcal{R}_0(\overline{\beta}, \theta^{\mathcal{S}}). \tag{42}$$

Putting together the two cases and simplifying, we obtain

$$\mathfrak{R}_{\mathrm{rob}}(\overline{\beta}; \Theta_{\mathrm{eq}}, M_{\mathrm{test}}) = \gamma'(C_{\mathrm{ker}} + \|R^\top \overline{\beta}\|_2)^2 + \mathcal{R}_{\mathrm{rob}}(\overline{\beta}; \theta^\mathcal{S}, M_{\mathrm{seen}})$$

$$= \gamma'(C_{\mathrm{ker}} + \|R^\top \overline{\beta}\|_2)^2 + \gamma(\beta^\mathcal{S} - \overline{\beta})^\top M_{\mathrm{seen}}(\beta^\mathcal{S} - \overline{\beta}) + \mathcal{R}_0(\overline{\beta}, \theta^\mathcal{S}), \tag{43}$$

where $\mathcal{R}_{\mathrm{rob}}(\overline{\beta}; \theta^\mathcal{S}, \gamma M_{\mathrm{seen}})$ is the robust risk of the estimator $\overline{\beta}$ w.r.t. the "identified" test shift $\gamma M$ and the identified model parameter $\theta^\mathcal{S}$, whereas $\mathcal{R}_0(\overline{\beta}, \theta^\mathcal{S})$ is the risk of $\overline{\beta}$ on the reference environment $e = 0$.

**Derivation of the lower bound for the worst-case robust risk.** Now that we have explicitly computed the worst-case robust risk, we devote ourselves to the computation of the lower bound for its best possible value

$$\inf_{\overline{\beta} \in \mathbb{R}^d} \mathfrak{R}_{\mathrm{rob}}(\overline{\beta}; \Theta_{\mathrm{eq}}, M_{\mathrm{test}}).$$

In this part, we will only consider the case $R \neq 0$, since the case $R = 0$ corresponds to the (discrete) anchor regression-like setting, where both the robust risk and its minimizer are uniquely identifiable, and computable from training data. We will distinguish between two cases.

**Case 1:** $\|R^\top \overline{\beta}\|_2 = 0$. In this case, $\overline{\beta}$ is fully located in the orthogonal complement of $R$, which consists of $\mathcal{S}$ and $\tilde{R}$ (the orthogonal complement or $R$ in $\mathcal{S}^\perp$). We will denote (the basis of) this subspace by $S_{\mathrm{tot}} = \mathcal{S} \oplus \tilde{R}$. Thus, $S_{\mathrm{tot}}$ is the "total" stable subspace consisting of identified directions in $\mathcal{S}$ and non-identified, but unperturbed directions $\tilde{R}$. We will parameterize $\overline{\beta}$ as $\overline{\beta} = S_{\mathrm{tot}} w$. Thus, we are looking to solve the optimization problem

$$\beta_{\Theta_{\mathrm{eq}}}^{\mathrm{rob}} = \arg\min_w (\beta^\mathcal{S} - S_{\mathrm{tot}} w)^\top (\gamma M_{\mathrm{seen}}^\top + \Sigma_\eta^\star)(\beta^\mathcal{S} - S_{\mathrm{tot}} w) + 2(\beta^\mathcal{S} - S_{\mathrm{tot}} w)^\top \Sigma_{\eta,\xi}^\mathcal{S} + (\sigma_\xi^\mathcal{S})^2.$$

Setting the gradient to zero yields the *asymptotic worst-case robust estimator*

$$\beta_{\Theta_{\mathrm{eq}}}^{\mathrm{rob}} = \beta^\mathcal{S} + S_{\mathrm{tot}}[S_{\mathrm{tot}}^\top(\gamma M_{\mathrm{seen}}^\top + \Sigma_\eta)S_{\mathrm{tot}}]^{-1} S_{\mathrm{tot}}^\top \Sigma_{\eta,\xi}^\mathcal{S}, \tag{44}$$

which corresponds to the loss value of

$$\mathfrak{R}_{\mathrm{rob}}(\beta_{\Theta_{\mathrm{eq}}}^{\mathrm{rob}}; \Theta_{\mathrm{eq}}, M_{\mathrm{test}}) = \gamma' C_{\mathrm{ker}}^2 + (\sigma_\xi^\mathcal{S})^2 - 2\Sigma_{\eta,\xi}^{\mathcal{S}\top} S_{\mathrm{tot}}[S_{\mathrm{tot}}^\top(\gamma M_{\mathrm{seen}}^\top + \Sigma_\eta)S_{\mathrm{tot}}]^{-1} S_{\mathrm{tot}}^\top \Sigma_{\eta,\xi}^\mathcal{S}.$$

As we observe, this quantity grows linearly in $\gamma'$. However, as $\gamma \to \infty$, the quantity *saturates* and is upper-bounded by $(\sigma_\xi^\mathcal{S})^2$.

**Case 2:** $\|R^\top \overline{\beta}\|_2 \neq 0$. Since for $\|R^\top \overline{\beta}\|_2 \neq 0$, the objective function is differentiable, we compute its gradient to be

$$\nabla \mathfrak{R}_{\mathrm{rob}}(\beta; \Theta_{\mathrm{eq}}, M_{\mathrm{test}}) = 2\gamma' RR^\top \beta / \|RR^\top \beta\| + 2\gamma' RR^\top \beta + \nabla \mathcal{R}_{\mathrm{rob}}(\beta; \theta^\mathcal{S}, \gamma M_{\mathrm{seen}})$$

$$= 2\gamma' RR^\top \beta / \|RR^\top \beta\| + 2\gamma' RR^\top \beta + 2(\Sigma_\eta^\star + \gamma M_{\mathrm{seen}})(\beta - \beta^\mathcal{S}) - 2\Sigma_{\eta,\xi}^\mathcal{S}.$$

This equation is, in general, not solvable w.r.t. $\beta$ in closed form. Instead, we provide the limit of the optimal value of the function when the strength of the unseen shifts is small, i.e. $\gamma' \to 0$. We know that for $\gamma' = 0$, the minimizer of the worst-case robust risk is given by the anchor estimator

$$\beta_{\mathrm{anchor}} = \beta^\mathcal{S} + (\Sigma_\eta^\star + \gamma M_{\mathrm{seen}})^{-1} \Sigma_{\eta,\xi}^\mathcal{S}.$$

Instead, we lower bound the non-differentiable term $2\gamma' C_{\mathrm{ker}} \|R^\top \beta\|$ by the scalar product $2\gamma' C_{\mathrm{ker}} \langle R^\top \beta, R^\top \beta_{\mathrm{anchor}} \rangle / \|\beta_{\mathrm{anchor}}\|$ and expect it to be tight for small $\gamma'$. After inserting this lower bound in Equation (41) we obtain the minimizer of the lower bound of form

$$\beta_{LB} = \beta^\mathcal{S} + (\Sigma_\eta^\star + \gamma M + \gamma' RR^\top)^{-1}(\Sigma_{\eta,\xi}^\mathcal{S} - \gamma' C_{\mathrm{ker}} RR^\top (\Sigma_\eta^\star + \gamma M)^{-1} \Sigma_{\eta,\xi}^\mathcal{S}).$$

We can now lower bound $\|RR^\top \beta_{LB}\|$ as

$$\|RR^\top \beta_{LB}\| \geq \|RR^\top (\Sigma_\eta^\star + \gamma M)^{-1} \Sigma_{\eta,\xi}^\star\| - \gamma' \cdot \mathrm{const}. \tag{45}$$

Thus, the $\gamma'$-rate of the worst-case robust risk of $\beta_{LB}$ is at least $\gamma'(C_{\mathrm{ker}} + \|RR^\top(\Sigma_\eta^\star + \gamma M)^{-1} \Sigma_{\eta,\xi}^\star\|)^2 + \mathcal{O}(\gamma'^2)$, from which the claim for small $\gamma'$ follows. For Section 3.2, the lower bound directly implies optimality of the worst-case robust risk of the anchor estimator when the strength of the unseen shifts $\gamma'$ is small. Additionally. if $\gamma = 0$, i.e. only unseen test shifts occur, we conclude that the OLS and anchor estimators have the same rates.

**Lower bound $\gamma'_{\text{th}}$ for $\gamma'$.** Finally, we want to derive a lower bound on the shift strength $\gamma'$ such that for all $\gamma' \geq \gamma'_{\text{th}}$ Case 1 of our proof is valid, i.e. it holds that $\beta^{\text{rob}}_{\Theta_{\text{eq}}}$ is given by the closed form "abstaining" estimator (44). For this, we find $\gamma'_{\text{th}}$ such that for all $\gamma' \geq \gamma'_{\text{th}}$ zero is contained in the subdifferential of$\mathfrak{R}_{\text{rob}}(\beta^{\text{rob}}_{\Theta_{\text{eq}}}; \Theta_{\text{eq}}, M_{\text{test}})$ at $\beta^{\text{rob}}_{\Theta_{\text{eq}}}$. Then the KKT conditions are met, and $\beta^{\text{rob}}_{\Theta_{\text{eq}}}$ is the unique minimizer of the worst-case robust risk due to strong convexity of the objective. We compute the subdifferential to be

$$S = \gamma' C_{\text{ker}} \{RR^\top \beta : \|\beta\|_2 \leq 1\} + \nabla \mathcal{R}_{\text{rob}}(\beta^{\text{rob}}_{\Theta_{\text{eq}}}; \theta^{\mathcal{S}}, \gamma M).$$

Since $\beta^{\text{rob}}_{\Theta_{\text{eq}}}$ is the minimizer of $\mathcal{R}_{\text{rob}}(\beta; \theta^{\mathcal{S}}, \gamma M_{\text{seen}})$ under the constraint $R^\top \beta = 0$, the gradient is zero in $R^\perp$ and it remains to show that

$$\|RR^\top \nabla \mathcal{R}_{\text{rob}}(\beta^{\text{rob}}_{\Theta_{\text{eq}}}; \theta^{\mathcal{S}}, \gamma M_{\text{seen}})\| \leq \gamma' C_{\text{ker}},$$

or

$$\gamma' \geq \|RR^\top \nabla \mathcal{R}_{\text{rob}}(\beta^{\text{rob}}_{\Theta_{\text{eq}}}; \theta^{\mathcal{S}}, \gamma M_{\text{seen}})\| / C_{\text{ker}}.$$

Via an upper bound on the projected gradient, we derive the stricter condition

$$\gamma' \geq \frac{\|RR^\top \Sigma^{\mathcal{S}}_{\eta, \xi}\|(1 + \kappa(\Sigma^\star_\eta))}{C_{\text{ker}}},$$

where $\kappa(\Sigma^\star_\eta)$ is the condition number of the covariance matrix.

### F.3 Proof of Corollary 3.2

To obtain a new formulation for the worst-case robust risk, we start with (46) and expand

$$\begin{aligned}
\mathfrak{R}_{\text{rob}}(\overline{\beta}; \Theta_{\text{eq}}, M_{\text{test}}) &= \gamma'(C_{\text{ker}} + \|R^\top \overline{\beta}\|_2)^2 + \gamma(\beta^{\mathcal{S}} - \overline{\beta})^\top M_{\text{anchor}}(\beta^{\mathcal{S}} - \overline{\beta}) + \mathcal{R}_0(\overline{\beta}, \theta^{\mathcal{S}}) \\
&= \gamma'(C_{\text{ker}} + \|R^\top \overline{\beta}\|_2)^2 + \gamma(\beta^{\mathcal{S}} - \overline{\beta})^\top M_{\text{anchor}}(\beta^{\mathcal{S}} - \overline{\beta}) \\
&\quad + (\beta^{\mathcal{S}} - \overline{\beta})^\top \Sigma^\star_\eta(\beta^{\mathcal{S}} - \overline{\beta}) + 2(\beta^{\mathcal{S}} - \beta)\Sigma^{\mathcal{S}}_{\eta, \xi} + (\sigma^{\mathcal{S}}_\xi)^2 \\
&= \gamma'(C_{\text{ker}} + \|R^\top \overline{\beta}\|_2)^2 + (\gamma - 1)(\beta^{\mathcal{S}} - \overline{\beta})^\top M_{\text{anchor}}(\beta^{\mathcal{S}} - \overline{\beta}) + \mathcal{R}(\beta, \mathbb{P}^{X,Y}_{\text{train}}),
\end{aligned}$$
$$(46)$$

where we have used that the pooled second moment of $X$ equals to $\Sigma^\star_\eta + \sum_e w_e(\mu_e \mu_e^\top) = \Sigma^\star_\eta + \gamma M_{\text{anchor}} - (\gamma - 1)M_{\text{anchor}}$. This reformulation shows that the worst-case robust risk is equal to the anchor population loss (cf. [41]) with an additional non-identifiability penalty term $\gamma'(C_{\text{ker}} + \|R^\top \overline{\beta}\|_2)^2$.

We now want to evaluate the rates of the anchor and OLS estimators in terms of the magnitude $\gamma'$ of unseen shift directions. We observe that only the non-identifiability term depends on $\gamma'$, whereas the second term only depends on $\gamma$. First, we compute the closed-form anchor regression estimator, which reads

$$\beta_{\text{anchor}} = \arg\min_{\beta \in \mathbb{R}^d} \mathcal{R}_{\text{rob}}(\beta, \theta^{\mathcal{S}}, \gamma M_{\text{anchor}}) = \beta^{\mathcal{S}} + (\Sigma^\star_\eta + \gamma M_{\text{anchor}})^{-1} \Sigma^{\mathcal{S}}_{\eta, \xi}. \tag{47}$$

Since $\beta_{\text{OLS}}$ equals to the anchor estimator with $\gamma = 1$, we obtain

$$\beta_{\text{OLS}} = \beta^{\mathcal{S}} + (\Sigma^\star_\eta + M_{\text{anchor}})^{-1} \Sigma^{\mathcal{S}}_{\eta, \xi}.$$

The claim of the corollary now follows by computing $\|RR^\top \beta_{\text{anchor}}\|$ and $\|RR^\top \beta_{\text{OLS}}\|$ and observing that the rest of the terms is constant on $\gamma'$.

Comparing to the lower bound (45) for the minimax quantity for the case of $\gamma' \to 0$, we observe that the anchor estimator is optimal (achieves the minimax rate) in the limit $\gamma' \to 0$. Additionally, if $\gamma = 0$ (only new shifts occur during test time), anchor and OLS have identical rates in $\gamma'$ and, in particular, OLS (corresponding to vanilla empirical risk minimization) is minimax-optimal in the limit of small unseen shifts.

