# OpenReview forum: "Achievable distributional robustness when the robust risk is only partially identified"
_NeurIPS.cc/2024/Conference — NeurIPS 2024 poster_

### Official Review · Reviewer_2y9v · 2024-07-10

**Soundness:** 3
**Presentation:** 3
**Contribution:** 3
**Rating:** 7
**Confidence:** 1

**Summary:**

This paper proposed a general framework of partially identifiable robustness to evaluate robustess in scenarios where the training distributions are not heterogeneous enough to identify the robust risk. They define 'the identifiable robust risk' and its correspondig minimax quantity. They show previous approaches achieve suboptimal robustness in this scenario.  Finally, they propose the empirical minimizer of the identifiable robust risk and show that it outperforms existing methods in finite-sample experiments.

**Strengths:**

The paper is crealy written. The idea of establishing partially identifiability to fill the gap between an "all-or-nothing" view on robustness in the SCM framework is interesting.

**Weaknesses:**

I am not an expert in this field so I cannot point out many weaknesses. Maybe one thing is the lack of motivation from real-world application, such as in what applications or when the proposed framework is usefull in practice?

**Questions:**

1. On line 135-137, the $M_{\text{test}}$ is defined as  $M_{\text{test}}=\gamma\Pi_{M}$. What's the reasoning behind this choice?

**Limitations:**

yes

---

> ### Author Rebuttal · Authors · 2024-08-07
>
> We are glad that the reviewer appreciated the main idea of our paper. In the following, we hope to clarify the real-world motivation for our study and some theoretical assumptions.
>
> **On real-world applications** The main motivation for our paper was expanding the existing assumptions in distribution shift literature to allow for a statement in a more realistic setting, particularly in safety-critical applications. We wanted to account for the fact that distribution shifts are mostly bounded in the real world and occur in “realistic” directions. Contrary to many existing results, in the real world, we rarely have access to enough training environments to achieve point identification of the “best” robust predictor. Instead, we aim to study the “best” model if only a few training environments are given, but the distribution shift (in potentially new directions) is not too large. In the following, we provide a **toy example**, motivated by medical applications, that illustrates our setting.
>
> Suppose we are conducting a long-term medical study, where data is collected from the same group of patients over the years to predict a health parameter $Y$, e.g., cholesterol level in the blood, from a group of covariates $X = (X_1, …, X_n)$ such as age, blood pressure, physical activity, resting pulse, BMI, etc. We are given data $(X^e, Y^e)$ for $e \in E_{train}$ from multiple past studies, where $\mathcal{E}_{train} = \{2010, 2015\}$ are the years in which the studies were conducted. We assume that the data $(X, Y)$ are generated by an underlying causal model in which not every variable is observed (confounded setting). In this example, we can identify the causal effect of the covariate $X_1$, age, on the cholesterol level since age has a mean shift of 5 years across the studies. Suppose, however, that the distribution of $X_2$, physical activity, remains relatively stable across the studies. The causal effect of $X_2$ on $Y$ is then not identifiable. We now want to train a model that generalizes best on the data $X^{2020}$ collected in $2020$. The age variable shifts again by 5 years; however, the physical activity variable now also shifts a bit (e.g., due to COVID-19), which is a shift we have not observed previously. In this situation, both the causal parameter and the robust predictor are only partially identifiable.
>
> **Assumption on $M_{test}$**: Often, practitioners will have some information on the strength and direction of the distribution shift (for instance, we might know that only the resting pulse will shift across studies, and at most by 20). This knowledge is what we tried to formalize in our assumption on $M_{test}$ (lines 134-137). The parameter gamma corresponds to the maximum strength of the shift, whereas the subspace $M$ corresponds to the expected direction of the shift (if one doesn’t know, one can take $M$ to be the whole covariate space). Equations (3) and (4) bound the second moment of the test distribution shift by $\gamma \Pi_{M}$, effectively bounding the mean and variance of the shift, as well as constraining it geometrically to the subspace $M$.

---

> > ### Comment · Reviewer_2y9v · 2024-08-11
> >
> > Thanks for the reply. It addressed my questions.

---

### Official Review · Reviewer_faJC · 2024-07-13

**Soundness:** 3
**Presentation:** 3
**Contribution:** 3
**Rating:** 7
**Confidence:** 3

**Summary:**

This paper investigates the optimal minimax risk of a robust predictor when the robustness set is partially observable, under a structural causal model with hidden confounders. By decomposing the test covariance matrix of latent parameters into a component spanned by the training distributions and the orthogonal component, it is shown that the robust predictor is only identifiable if the test shifts are in the direction of the training shifts. While for unobserved test shifts, the best achievable minimax risk grows linearly with shift scale. The theory is applied to show sub-optimality of OLS and anchor regression with partially observable test shifts.

**Strengths:**

I am not familiar with the field of causal inference, but it appears from the paper that it has made contributions as the first result for distributional robustness or invariant causal prediction when the robustness set is partially identifiable. Particularly, it shows that infinite robustness is impossible under this scenario, and finite robustness methods can show performances reduced to ERM. This echoes empirical evidence and provides a possible theoretical explanation for reported failure of distributional robustness methods under wild environments.

**Weaknesses:**

The structural causal model in Eq 2 and the resulting explicit solution in Eq 6 shows that the model is biased even without distribution shifts. To see this, take $\gamma =0$ in Eq 6 and the predictor does not reduce to $\beta^\star$. The model is unbiased only when the cross covariance between $\eta$ and $\xi$ vanishes, which implies no hidden confounder and only covariate shift. In contrast, the classic approach for invariant causal prediction [1] produces unbiased estimation beyond covariate shift.

[1] Arjovsky, M., Bottou, L., Gulrajani, I., & Lopez-Paz, D. (2019). Invariant risk minimization. arXiv preprint arXiv:1907.02893.

**Questions:**

1. Typo: L137, the sub-space.
2. In Fig 1, bidirectional edges don't make sense for causal graphs, and it's not explained either. Moreover, is there exact equivalence between Fig 1 and the SCM in Eq 2?

**Limitations:**

The author has addressed limitations with regarding to the model, e.g., linear structure and additive noises.

---

> ### Author Rebuttal · Authors · 2024-08-07
>
> We thank the reviewer for appreciating the contribution of the paper and precisely summarizing its main idea.
>
> **Bias of the structural causal model** The reviewer is correct that the causal effect estimate would be biased when the cross-correlation between the noises $\xi$ and $\eta$ is not 0. We would now like to argue that this is a feature rather than a bug. First, we would like to clarify that this paper considers the prediction performance, or MSE, during test time. Therefore, having a biased causal effect estimate does not necessarily affect the test performance, which is still optimal for $\gamma=0$ (no shift). Furthermore, allowing for latent confounding (expressed via cross-correlation of $\eta$ and $\xi$) only makes our model more general and the problem much more challenging to solve. The presence of confounding creates a tradeoff between predictive power and robustness (see, e.g., [2]).
> Furthermore, we would like to highlight that even IRM might not always produce unbiased estimates of the causal coefficient. For example, in [1], the synthetic data experiment shows that IRM and OLS have a bias of similar magnitude in a confounded setting with homoskedastic noise.
>
> **Q2 (bidirectional edges)** Normally, a bidirectional edge $X \leftrightarrow Y$ between two nodes indicates latent confounding. In Figure 1, we explicitly list the latent confounder $H$, resulting in the notation $X \leftrightarrow H \leftrightarrow Y$,  to express that the latent confounder can be both an ancestor and descendant of $X$ and/or $Y$. More precisely, we allow for the following scenarios: $X  \rightarrow H \rightarrow Y$, $X \leftarrow H \rightarrow Y$, $X \rightarrow H \leftarrow Y$ (the fourth one is excluded due to the acyclicity constraint).
>
> **Q2 (equivalence of Figure 1 and the SCM)**  To see the equivalence between the SCM and DAG, it is helpful to consider the three configurations discussed before. When $X \leftarrow H \rightarrow Y$, the causal coefficient $\beta^\star$ equals the weight of the path $X \rightarrow Y$. In this setting, $\eta$ and $\xi$ are correlated via $H$. When $X \rightarrow H \rightarrow Y$, the causal coefficient $\beta^\star$ equals the sum of the weights along the paths $X \rightarrow Y$ and $X \rightarrow H \rightarrow Y$. In this setting, $\eta$ and $\xi$ are independent. When $X \rightarrow H \leftarrow Y$, the causal coefficient $\beta^\star$ equals the weight of the path $X \rightarrow Y$. In this setting, $\eta$ and $\xi$ are independent.
>
> [1]: Arjovsky M, Bottou L, Gulrajani I, Lopez-Paz D. Invariant risk minimization. arXiv preprint arXiv:1907.02893. 2019 Jul 5.
>
> [2]: Rothenhäusler D, Meinshausen N, Bühlmann P, Peters J. Anchor regression: Heterogeneous data meet causality. Journal of the Royal Statistical Society Series B: Statistical Methodology. 2021 Apr;83(2):215-46.

---

> > ### Comment · Reviewer_faJC · 2024-08-07
> >
> > I acknowledge and thank the author for the response. In my review, my concern over the usefulness of the structural causal model is addressed. I recognize that the non-existence of an unbiased causal effect estimator for infinite robustness is expected where bounded distribution shift is considered, which is also part of the contribution. Therefore, I raise my score from 6 (weak accept) to 7 (accept).

---

### Official Review · Reviewer_iX3g · 2024-07-14

**Soundness:** 3
**Presentation:** 3
**Contribution:** 3
**Rating:** 7
**Confidence:** 4

**Summary:**

This paper proposes a new framework for distributionally robustness under the linear causal setting. Specifically, the authors minimize the so-called identifiable robust risk, which corresponds to the maximum of the robust risk for parameters in the observationally equivalent set. Under such partially identifiable robustness framework, they discuss the lower bound of the risk and show that some estimated identifiable robust predictor can achieve that lower bound risk, where the corresponding empirical version can approximate the lower bound risk well. They also validate the better performance with finite sample numerical experiments.

**Strengths:**

-	The motivating illustration and results are very clear;
-	The direction the authors study is interesting in bridging the structure and non-structure of distributional robustness.

**Weaknesses:**

-	Some paragraph in the training and testing data part can be reorganized as some formal assumptions, such as the assumption of M_{test} and linear assumption as well as the discussion in Section 3.2 in terms of the definition of S and M. This gives a clear picture on what some key conditions (and possible relaxations) are in this paper.
-	There are a few missing details (see Questions as follows), which I hope the authors can explain them a bit more.

**Questions:**

-	Can the authors highlight a bit more on the (approximated) real-world example when the predictor is partially identifiable? Since it may be a bit unfamiliar to the general DRO audience for these notations and their relevance in practice.
-	Can the authors elaborate more on how the empirical estimations for the space of the training shift direction $\hat S$ and $\hat R$ are computed, as compared with the current version in Appendix D? From my preliminary understanding, this space determination is important for subsequent estimation.
-	The potential utility of **active intervention selection** is also interesting, and it seems aligned with some recent relevant work on distribution shifts. For example, in causal explanation [1] and incorporating specific features in improving distributional robustness [2]. I am wondering if the authors can provide more details on how their own intervention is and the connections with these existing literature, which can be potentially incorporated in the main body and appendix.

[1] Quintas-Martinez, Victor, et al. "Multiply-Robust Causal Change Attribution." arXiv preprint arXiv:2404.08839 (2024).

[2] Liu, Jiashuo, et al. "On the Need for a Modeling Language Describing Distribution Shifts: Illustrations on Tabular Datasets." arXiv preprint arXiv:2307.05284 (2023).

**Limitations:**

The authors discuss the main limitation of this paper, which I think is reasonable compared with existing literature.

---

> ### Author Rebuttal · Authors · 2024-08-07
>
> We thank the reviewer for pointing out the novelty of the direction of our study. We are happy to fill in on the missing details below:
>
> **Q1 (real-world example of partial identifiability)** Indeed we can give a toy example that illustrates our abstract setting in Section 3.1: suppose that we are conducting a long-term medical study, where data is collected from the same group of patients over the years to predict a health parameter $Y$, e.g., cholesterol level in blood, from a group of covariates $X = (X_1, …, X_n)$ such as age, blood pressure, physical activity, resting pulse, BMI etc. We assume that the data $(X, Y)$ are generated by an underlying causal model, in which not every variable is observed (i.e. there might exist latent confounding). During training time, we are given data $(X^e, Y^e)$ for $e \in E_{train}$ from multiple past studies, where $E_{train} = \{2010, 2015\}$ are the years in which the studies were conducted. We now want to train a model which generalizes best on data $X^{2020}$ collected in 2020.
>
> In such a setting, we can expect that the distribution shift between the data in 2010, 2015 and 2020 is not entirely arbitrary: for example, the covariates such as age ($X_1$) might shift similarly between 2010 to 2015 and 2015 to 2020, whereas other covariates such as physical activity ($X_2$), BMI ($X_3$) or resting pulse ($X_4$) may exhibit unseen, but bounded shifts (e.g. due to a disease outbreak such as Covid). This is an example of a structured distribution shift. We observe that we can identify the causal parameter $\beta^\star$ in the direction of $X_1$, since there is a shift of age across the training environments. However, the causal parameter is not identifiable for $X_2$ if the distribution of physical activity has not shifted in previous studies. This renders the causal parameter, and (since the test data shifts in the direction of $X_2$) the robust risk of the problem, partially identifiable. In this setting, only $X_1$ is guaranteed to give an invariant prediction, but an estimator which just uses the age might not have enough predictive power. Instead, we suggest to also utilize the spurious correlations between $X_2,...,X_4$ and $Y$ to predict the target variable, but penalize the predictor in their directions based on the strength of the expected test shift (e.g., we do not expect the average resting pulse to shift by more than 10).
>
> **Q2 (empirical estimation of S and R)** Indeed, empirical estimation of the subspaces $S$ and $R$ is an important part of the practical application of our method. We will add to the manuscript that the space $S$ can be computed from the second moments of the observed distribution via $S := \text{span} \cup_{e \in E_{train}} [(\Sigma_e - \Sigma_0) + (\mu_e - \mu_0)(\mu_e - \mu_0)^\top]$, and can be estimated via empirical means and covariances of the training distributions. Results on the consistency of estimation of the eigenvectors when the dimension is fixed and eigenvalues are finite are given, for example, by [1]. For the test shift, there’s either prior information on their direction available via a subspace $M$, which we can incorporate by setting $R = \Pi_{S^{\perp}} M$, or, if not, we may take the conservative estimate of $R$ being the orthogonal complement of $S$. The downside of this choice is that the robustness requirements might be more restrictive than necessary.
>
> **Q3a (active intervention selection)** Thanks a lot for this comment. Although we plan to have a longer elaboration of this topic in the full manuscript, we briefly discuss here how our approach might be utilized for active intervention selection. In Equation (31), the identifiable robust risk is given by a supremum over the possible true causal parameters from the observationally equivalent set. The argmax of the supremum (which depends on the estimator) utilizes partial identifiability to find the “most adversarial” direction for the test shift, along which the given estimator will suffer the highest test risk. One can actively sample the next dataset by performing an intervention along this direction, which will maximally decrease the identifiable robust risk (31) of a given estimator (e.g., OLS). This procedure can then be repeated.
>
> **Q3b connections to listed literature** In [2] (causal change attribution), the main difference seems to be that the causal graph is known and the objective is, instead of robust prediction, causal change attribution. A conceptual similarity seems to be that [2] allows for recovery of the target parameter under partial misspecification. However, in our framework this would mean that although some components of the model might be misspecified (or, in our words, underidentified), the target parameter is still identifiable – thus, we would call the setting in [2] identifiable under the listed assumptions, similarly to the case of anchor regression. [3] is fairly related to our work motivation-wise: our model allows for $Y|X$-shifts in addition to $X$-shifts, consistent with the observation that $Y|X$-shifts frequently occur in tabular data and cannot be neglected.
>
>
> [1]: Anderson TW. Asymptotic theory for principal component analysis. The Annals of Mathematical Statistics. 1963 Mar;34(1):122-48.
>
> [2]: Quintas-Martinez V, Bahadori MT, Santiago E, Mu J, Janzing D, Heckerman D. Multiply-Robust Causal Change Attribution. arXiv preprint arXiv:2404.08839. 2024 Apr 12.
>
> [3]: Liu J, Wang T, Cui P, Namkoong H. On the need for a language describing distribution shifts: Illustrations on tabular datasets. Advances in Neural Information Processing Systems. 2024 Feb 13;36.

---

> > ### Comment · Reviewer_iX3g · 2024-08-09
> >
> > I acknowledge and thank the authors for their detailed response. Therefore, I keep my evaluation towards the paper.
> >
> > Specificially, I am quite interested in the active intervention part authors mention and agree with what the authors say in the rebuttal with respect to its connection with the active learning side. Therefore, I am eager to see their revised version. Btw, for [3] (i.e. https://arxiv.org/pdf/2307.05284), I am pointing out their intervention part (i.e., Section 5 in their latest arXiv version instead of the conference version), which also discusses some potential feature-based interventions and can be of separate utility to the authors.

---

### Official Review · Reviewer_PdW8 · 2024-07-29

**Soundness:** 4
**Presentation:** 3
**Contribution:** 3
**Rating:** 6
**Confidence:** 4

**Summary:**

The paper studies a linear Structural Causal Model (SCM) for prediction under distribution shifts due to an an additive term to covariates that changes at test time, and an unobserved confounder between the covariates and the label. The key difference between the proposed analysis and those presented in other works (e.g. those in anchor regression, invariant causal prediction etc.) is that the shift is bounded in its strength. Another difference is that the optimal robust predictor w.r.t the entire uncertainty set is not identifiable. Hence, instead of the optimal robust error, the paper studies the optimal error that can be achieved from observable data. This corresponds to the robust error over an uncertainty set that is generated by all SCM parameters that could have produced the training data, which is in general a larger set than the uncertainty set we are interested in (i.e. of bounded additive shifts to the covariates).

Once the problems is set up, a parametric form for the set of parameters that can generate the training data is derived, along with the corresponding set of possible robust predictors. Then under a mild assumption on boundedness of the ground truth regression parameters, the following are derived: 1) a closed form is derived for the robust loss in the case where test distributions only induce shifts in directions that have been observed at training. The loss grows linearly with the strength of the shift; 2) a lower bound on the achievable risk from observed data, which is tight (and thus can be learned from observed data) for large enough shifts.

Further analysis and simulations with Gaussian data are done to compare the risk of the derived estimator with that of ERM and anchor regression (which does not exploit the boundedness of shifts to achieve better prediction). The results verify that the risk obtained by the proposed empirical estimator are close the lower bound given in the theorem, and outperform the two baselines as shifts grow large.

**Strengths:**

* Overall I enjoyed reading the paper, it is clear and written with care. Generally, I also like the direction of formalizing bounded shifts and unidentifiable settings in detail. Finally, the analysis is well performed and easy to follow.

* The work is original in formally analyzing bounded distribution shifts, where even in population the optimal robust predictor might still have non-zero projection on directions that shift at test time. \
Small note on this: I think that for finite sample guarantees boundedness assumptions on the strength of the shift must be made, and they are made in works that give sample complexity results. I believe the reason is that with unbounded shifts, any small weight on a shifting feature can be magnified unboundedly to yield a large robust error. With finite samples, it is usually impossible to guarantee strictly zero weights on the shifting directions. Therefore it might be worthwhile clarifying that considering the population loss is an important component for the analysis.

* Beyond the points mentioned above, the bounds on the robust risk, closed form solutions, and the formalism used in the work, can be useful for future theoretical analyses. In terms of practical significance, the approach derived from these results to explicitly account for bounded shifts might be useful in the future, if it is generalized beyond linear models.

**Weaknesses:**

* As mentioned above and the authors mention in describing the limitations of their work, it is restricted to linear models. Intuitively, a considerable challenge in learning robust models is to identify the ``directions” that shift between domain. Under linear models it is rather straightforward, and the method proposed in the paper depends on linearity of the model in order to find these directions. This is unlike some other methods in the domain generalization literature, where certain formal results are provided on linear models but the methods can easily be tested in the non-linear case.
*  Even within the realm of linear models, I am not throughly convinced that the method is useful in real world problems. To make this more convincing, it might have been nice to run simulations over non-synthetic data and some more variants of methods. E.g. using several methods that learn invariant models when hyper-parameter tuning is performed with the objective proposed in this work may be of interest. That is to see whether boundedness of the shift should be taken into account during training, or is it enough to simply use it for model selection. Yet the most significant drawback is still of course, the limitation to linear models.

Some other/smaller comments:
* In line 60 prior work is cited to claim "that even minor violations of the identifiability assumptions can cause invariance based methods to perform equally or worse than ERM". However, I am not sure that the failures portrayed in these works are strictly due to identifiability violations. At least not the ones alluded to in this paper, which is the lack of heterogeneity in the training environments. In Kamath et al. 21, the predictor is identifiable but the problem is specifically in the IRMv1 objective, see also [1] who point out an objective that solves this issue. In Rosenfeld et al. 20, only one failure is due to not having enough environments (and arguably, this is since they consider the uncertainty set that includes shifts in all directions. I will also touch on this in the next minor comment), and the second one is due to non-linearity.
* Regarding analysis under lack of heterogeneity or unidentifiable optimal robust predictor: I think that the part of the analysis that touches upon the insufficient heterogeneity of the environments in order to identify the robustness set/robust predictor can be slightly reframed. If I am not mistaken, even in works that give results about identifiable robustness sets (e.g. when the number of environments is linear in the number of shifting features, or stronger ones like [2]), it is most likely possible to draw guarantees about robust risks, but only with respect to a smaller uncertainty set which is restricted to dimensions that shifted between the training environments. That is since the methods still enforce constraints which restrict weights in certain directions. While it is true that most of these prior works did not consider unidentifiable uncertainty sets, it might be related to choice of presentation, and not strictly because the methods are technically limited in that sense. Further, in lines 168-169 it is claimed that “prior work only considers scenarios where the robustness set and hence also the robust prediction model are still identifiable”. I am not sure this is entirely true. There are different formalisms that were considered for spurious correlations, such as those in [3]. In their setting, when the association is not what they call “purely spurious”, then the robust predictor cannot necessarily be recovered. Also [4] study a similar setting where due to similar reasons no guarantee can be given on identifying the robust predictor, but only a lower bound on the error is given.
* It might be worth mentioning in lines 112-113 that in case $H$ in figure 1 is a selection variable (I assume it can be since both edges are bi-directional) then the shift reduces to covariate shift, if I am not mistaken.
* Personally, I found the notation in the introduction which uses $\mathrm{shift}\in{\mathrm{shift set}}$ etc. somewhat redundant and confusing. The more formal notation in section 2 onwards was much easier to understand, so in my opinion it might be worth considering to go straight ahead into that notation.

[1] Wald Y, Feder A, Greenfeld D, Shalit U. On calibration and out-of-domain generalization. Advances in neural information processing systems. 2021 Dec 6;34:2215-27.

[2] Chen Y, Rosenfeld E, Sellke M, Ma T, Risteski A. Iterative feature matching: Toward provable domain generalization with logarithmic environments. Advances in Neural Information Processing Systems. 2022 Dec 6;35:1725-36.

[3] Veitch V, D'Amour A, Yadlowsky S, Eisenstein J. Counterfactual invariance to spurious correlations in text classification. Advances in neural information processing systems. 2021 Dec 6;34:16196-208.

[4] Puli AM, Zhang LH, Oermann EK, Ranganath R. Out-of-distribution Generalization in the Presence of Nuisance-Induced Spurious Correlations. In International Conference on Learning Representations.

**Questions:**

* It could’ve been nice to derive upper bounds on the robust loss in addition to the lower bounds proved in the theorem. That is, assuming these bounds can be calculated from observable data, I’d imagine that most practitioners would be more interested in an upper bound as it gives a strong guarantee on the risk worst possible risk. Do you think this is a reasonable goal, and do you perhaps have any insights on this?
* Could you perhaps discuss the differences between the empirical optimization problem you derived in the paper and those derived in prior work? For instance, looking at eq. 19 in appendix D, it seems like there is a term that penalizes $\|\| \hat{S}^\top(\hat{\beta}^{\mathcal{S}} - \hat{\beta}) \|\|$ which I read as the weights in the “invariant directions” should be the same across the environments. This seems quite similar to ICP, IRM etc. Then the other term limits the projection on the shifting directions, but it might be useful for readers to have a short discussion on the conceptual similarities between this and penalized versions of other methods. Also, there might be a typo in that equation, $\mathrm{arg}\min_{\beta\in{\mathbb{R}^d}}$ should be $\mathrm{arg}\min_{\hat{\beta}\in{\mathbb{R}^d}}$ instead.

**Limitations:**

The authors have adequately discussed limitations.

---

> ### Author Rebuttal · Authors · 2024-08-07
>
> We thank the reviewer for appreciating the overall direction and clarity of our paper, as well as originality of our work. We are grateful for the constructive comments and will incorporate the minor points in the revised version of the paper. We now respond to some of the major points and questions.
>
> **Linearity of the model** We acknowledge that understanding which shift directions are “learned” and which are “unobserved” during training time for more general, nonlinear models is an important direction for future work. We further recognize the linearity of our setting as a limitation. We would like to clarify that the main contribution of our paper is not primarily to propose a new method for domain generalization (that can then also be tested in non-linear settings), but to introduce and make a first step towards quantifying the limits of robustness in a partially identifiable setting. On this note, we expect that the results and intuition developed in this paper for the linear case can be utilized beyond linear models, since realistic distribution shifts can be often reduced to linear shifts in a lower-dimensional latent space via a suitable parametric or non-linear map [5, 6].
>
> **Evaluation of the method in real-world settings** Even though the method is not necessarily the focus of our paper, we agree that our work could benefit from a more thorough evaluation on real-world data. To this end, in the attached PDF, we present preliminary results of experiments on real-world single-cell gene expression data [7]. The dataset consists of single-cell observations of over 622 genes from observational and several interventional environments, arising through a knockdown of each unique gene. We pre-select always-active genes in the observational setting, resulting in a smaller dataset of 28 genes. We measure the performance of our method, the identifiable robust predictor (Rob-ID), compared to other algorithms [10, 11, 12], as follows. We select each gene once as the target variable $Y$ and select the three genes most strongly correlated with $Y$ (using Lasso), resulting in a smaller dataset.
> Given this dataset, we test on all combinations of training (observational + one shift) and test environments (other shift).
> We train the algorithms on the training environments and evaluate them on the test environments using mean square error (MSE). The results in Figures 2(a) and 2(b) indicate that Rob-ID outperforms existing distributional robustness methods, particularly for larger shifts in new directions.
>
> **Failure cases of IRM as motivation for our work** We agree that while the cited papers include non-identifiability as failure cases (Kamath et al. ‘21 Section 4 and Rosenfeld et al. ‘20), they also discuss other possible reasons for failure. We can clarify this in the revised version by adding that multiple failure cases have been discussed in the literature (citing these papers), one of which is non-identifiability, the focus of this work.
>
> **Application of existing work for partially identifiable case** We agree that prior methods can be applied and evaluated in the non-identifiable setting even though prior analysis so far has focused on the case where the robust risk was identifiable (including your described case when the ”uncertainty set which is restricted to dimensions that shifted between the training environments”). In fact, one of the goals of our paper was exactly to evaluate prior methods in this new setting.
>
> **Existing formalisms for spurious correlations**
> We thank the reviewer for pointing us to relevant and interesting works [3] and [4]. Although these works describe cases where the optimal robust predictor cannot be recovered, they seem to be binary negative statements in that they do not provide a quantification of this failure nor a proposed method for robustness in this failure case.
>
> **Q1 (upper bounds)** In our result, for $\gamma > \gamma_{th}$, the value of the inf sup is exact, thus, in Theorem 3.1, Case (b) line 2 it is both a lower and an upper bound (we will clarify this in the main text). For small $\gamma$, the loss of the anchor regression estimator is a tight upper bound – which for the practitioner means that if the shifts in unknown directions are expected to be very small, the anchor estimator is optimal.
>
> **Q2 (connections to invariance objectives and regularizers)** Thanks for the suggestion, we will include a more intuitive interpretation of the individual terms in the revised manuscript. In particular, we will discuss the population objective (13) that corresponds to the empirical objective in (19) when we replace $S$, $\beta^S$, $R$ by their empirical estimates. The objective  consists of three parts: 1) loss on the reference environment (without any distribution shift), 2) the term $||S^\top (\beta^S - \beta)||$ that “aligns” $\beta$ in the directions of the true causal parameter $\beta^\star$ projected on S, and 3) the term $(C_{ker} + || R^\top \beta ||_2)^2$  that shrinks $\beta$ in the directions unseen during training. Since $\beta^\star$ is the optimal invariant predictor (although unknown), the second term can be interpreted as aligning the estimator towards the true invariant predictor along the observed shifts (and hence corresponds to inducing invariance across training environments, spiritually corresponding to the invariant literature as the reviewer pointed out).
>
> For citations, please refer to the general rebuttal out of space reasons.

---

> > ### Comment · Reviewer_PdW8 · 2024-08-10
> >
> > Thank you for addressing the comments and questions, I also appreciate the additional empirical results.
> >
> > I see the point in most of the replies, perhaps one small comment is that [3, 4] indeed provide some binary negative statements, but [4] also give a form of a guarantee (an estimator that has better-than-random worst case performance, and is optimal in a restricted manner). I agree that it is a different flavor of guarantee and analysis than the one given in this paper. Hence my comment is just meant to suggest a slight adjustment to the framing of the contribution, not to devalue it in any way.
> >
> > I have raised my score following the in-depth response.

---

### Author Rebuttal · Authors · 2024-08-07

We express sincere gratitude to all reviewers for their detailed reviews. It is encouraging to hear that the reviewers appreciate the novelty and overall direction of our work, especially its attempt to formalize partially identifiable robustness.

We also appreciate constructive feedback and  we have carefully addressed and answered all points raised by the reviewers in the revised manuscript. For convenience, we summarize some important points that were mentioned by several reviewers and elaborate on some of them in more detail in the individual rebuttals.

- **Evaluation of the method on real-world data**: In the answer to Reviewer PdW8, we expand the empirical evaluation of our method to real-world single-cell gene expression data [7] – please see the attached PDF for preliminary results.
- **Meaning of “partial identifiability”**: It seemed that there might have been some remaining confusion about the way we use the term “partial identifiability”, also in the context of previous work.  We would like to stress that in this paper, we focus on partial identifiability of the *robust risk* that arises through the non-identifiability of the causal/model parameters. In particular, we distinguish between partial identifiability of the robust risk (not considered in prior work) and partial identifiability of the causal parameter (considered in prior work, e.g. [10]). For example, existing literature [10] often considers cases where, although the model parameters are not fully identifiable, the robust predictor can still be computed. In the language in our paper, we call such a setting fully instead of partially identifiable, since the robust risk and its minimizer can be computed from data, as opposed to our case, where neither the model nor the robust risk can be identified.
- **Relation to invariance-based literature**: In the answer to Reviewer PdW8, we explain how existing literature (e.g., [14,15]) on the failure of invariance-based methods (such as Invariant Risk Minimization) can be related to lack of identifiability and thus motivates our study.
- **Real-world examples/motivation**: In the answers to Reviewers iX3g and 2y9v, we describe a motivating toy example based on medical data where the robust predictor is partially identifiable.

Here we add the references for this general response and the rebuttal to Reviewer PdW8:

[1] Wald Y, Feder A, Greenfeld D, Shalit U. On calibration and out-of-domain generalization. Neurips 2021

[2] Chen Y, Rosenfeld E, Sellke M, Ma T, Risteski A. Iterative feature matching: Toward provable domain generalization with logarithmic environments. Neurips 2022

[3] Veitch V, D'Amour A, Yadlowsky S, Eisenstein J. Counterfactual invariance to spurious correlations in text classification. NeurIPS 2021

[4] Puli AM, Zhang LH, Oermann EK, Ranganath R. Out-of-distribution Generalization in the Presence of Nuisance-Induced Spurious Correlations. In ICLR 2021

[5]: Thams N, Oberst M, Sontag D. Evaluating robustness to dataset shift via parametric robustness sets. NeurIPS. 2022

[6]: Buchholz S, Rajendran G, Rosenfeld E, Aragam B, Schölkopf B, Ravikumar P. Learning linear causal representations from interventions under general nonlinear mixing. NeurIPS 2023

[7]: Chevalley M, Roohani Y, Mehrjou A, Leskovec J, Schwab P. Causalbench: A large-scale benchmark for network inference from single-cell perturbation data. arXiv preprint 2022

[9]: Schultheiss C, Bühlmann P. Assessing the overall and partial causal well-specification of nonlinear additive noise models. JMLR 2024

[10]: Rothenhäusler D, Meinshausen N, Bühlmann P, Peters J. Anchor regression: Heterogeneous data meet causality. JRSSB. 2021

[11]: Shen X, Bühlmann P, Taeb A. Causality-oriented robustness: exploiting general additive interventions. arXiv preprint 2023

[12]: Peters J, Bühlmann P, Meinshausen N. Causal inference by using invariant prediction: identification and confidence intervals. JRSSB 2023

[13]: Arjovsky M, Bottou L, Gulrajani I, Lopez-Paz D. Invariant risk minimization. arXiv preprint 2019

[14]: Rosenfeld E, Ravikumar P, Risteski A. The risks of invariant risk minimization. arXiv preprint 2020

[15]: Kamath P, Tangella A, Sutherland D, Srebro N. Does invariant risk minimization capture invariance?. AISTATS 2021.

---

### Decision · Program_Chairs · 2024-09-25

**Decision:**

Accept (poster)

**Comment:**

The paper received positive reviews, with the reviewers appreciating the novelty of the results and the quality of the presentation. Consequently, I recommend acceptance of this paper. When preparing the camera-ready revision, please pay close attention to the reviewers’ feedback and include the additional explanations provided during the rebuttal period.